# Many-molecule encapsulation by an icosahedral shell

**Jason D Perlmutter[†], Farzaneh Mohajerani[†], Michael F Hagan\***

Martin Fisher School of Physics, Brandeis University, Waltham, United States

**Abstract** We computationally study how an icosahedral shell assembles around hundreds of molecules. Such a process occurs during the formation of the carboxysome, a bacterial microcompartment that assembles around many copies of the enzymes ribulose 1,5-bisphosphate carboxylase/ oxygenase and carbonic anhydrase to facilitate carbon fixation in cyanobacteria. Our simulations identify two classes of assembly pathways leading to encapsulation of many-molecule cargoes. In one, shell assembly proceeds concomitantly with cargo condensation. In the other, the cargo first forms a dense globule; then, shell proteins assemble around and bud from the condensed cargo complex. Although the model is simplified, the simulations predict intermediates and closure mechanisms not accessible in experiments, and show how assembly can be tuned between these two pathways by modulating protein interactions. In addition to elucidating assembly pathways and critical control parameters for microcompartment assembly, our results may guide the reengineering of viruses as nanoreactors that self-assemble around their reactants.

**\*For correspondence:** hagan@brandeis.edu

[†]These authors contributed equally to this work

**Competing interests:** The authors declare that no competing interests exist.

## Introduction

Encapsulation is a hallmark of biology. A cell must co-localize high concentrations of enzymes and reactants to perform the reactions that sustain life, and it must safely store genetic material to ensure long-term viability. While lipid-based organelles primarily fulfill these functions in eukaryotes, self-assembling protein shells take the lead in simpler organisms. For example, viruses surround their genomes with a protein capsid, while bacteria use large icosahedral shells known as bacterial microcompartments (BMCs) to sequester the enzymes and reactions responsible for particular metabolic pathways (*Kerfeld et al., 2010*; *Axen et al., 2014*; *Shively et al., 1998*; *Bobik et al., 1999*; *Erbilgin et al., 2014*; *Petit et al., 2013*; *Price and Badger, 1991*; *Shively et al., 1973*; *Shively et al., 1973*; *Kerfeld and Erbilgin, 2015*). Within diverse bacteria, BMC functions have been linked to bacterial growth, carbon fixation, symbiosis, or pathogenesis (*Kerfeld and Erbilgin, 2015*). Other protein-based compartments are found in bacteria and archea (*e.g.* encapsulins (*Sutter et al., 2008*) and gas vesicles (*Pfeifer, 2012*; *Sutter et al., 2008*)) and even eukaryotes (*e.g.* vault particles (*Kickhoefer et al., 1998*)), while some viruses may assemble around lipidic globules (*Lindenbach and Rice, 2013*; *Faustino et al., 2014*). Thus, understanding the factors that control microcompartment assembly and encapsulation is a central question in modern cell biology. From the perspectives of synthetic biology and nanoscience, there is great interest in reengineering BMCs or viruses as nanoreactors that spontaneously encapsulate enzymes and reagents in vitro (*e.g. Luque et al., 2014*; *Douglas and Young, 1998*; *Rurup et al., 2014*; *Patterson et al., 2014*; *Patterson et al., 2012*; *Zhu et al., 2014*; *Rhee et al., 2011*; *Rurup et al., 2014*; *Wörsdörfer et al., 2012*; *Comas-Garcia et al., 2014*), or as customizable organelles that assemble around a programmable set of core enzymes in vivo, introducing capabilities such as carbon fixation or biofuel production into bacteria or other organisms (*e.g. Kerfeld and Erbilgin, 2015*; *Bonacci et al., 2012*; *Parsons et al., 2010*; *Choudhary et al., 2012*; *Lassila et al., 2014*). However, the principles

**eLife digest** Bacterial microcompartments are protein shells that are found inside bacteria and enclose enzymes and other chemicals required for certain biological reactions. For example, the carboxysome is a type of microcompartment that enables the bacteria to convert the products of photosynthesis into sugars. During the formation of a microcompartment, the outer protein shell assembles around hundreds of enzymes and chemicals. This formation process is tightly controlled and involves multiple interactions between the shell proteins and the cargo – the enzymes and other reaction ingredients – they will enclose. Understanding how to control which enzymes are encapsulated within microcompartments could help researchers to re-engineer the microcompartments so that they contain drugs or other useful products.

Recent studies have used microscopy to visualize how microcompartments are assembled. However, most of the intermediate structures that form during assembly are too small and short-lived to be seen. It has therefore not been possible to explore in detail how shell proteins collect the necessary cargo and then assemble into an ordered shell with the cargo on the inside. Experiments alone are probably not enough to understand the process, especially since microcompartment assembly can currently only be studied within live cells or cellular extract. Within these complex environments it is difficult to determine the effect of any individual factor on the overall assembly process.

Perlmutter, Mohajerani and Hagan have now taken a different approach by developing computational and theoretical models to explore how microcompartments assemble. Computer simulations showed that microcompartments could assemble by two pathways. In one pathway, the protein shell and cargo coalesce at the same time. In the other pathway, the cargo molecules first assemble into a large disordered complex, with the shell proteins attached on the outside. The shell proteins then assemble, carving out a piece of the cargo complex. The simulations showed that many factors affect how the shell assembles, such as the strengths of the interactions between the shell proteins and the cargo. They also identified a factor that controls how much cargo ends up inside the assembled shell.

Perlmutter, Mohajerani and Hagan found that, in addition to revealing how microcompartments may assemble within their natural setting, the simulations provided guidance on how to re-engineer microcompartments to assemble around other components. This would enable researchers to create customizable compartments that self-assemble within bacteria or other host organisms, for example to carry out carbon fixation or make biofuels.

A future challenge will be to investigate other aspects of microcompartment assembly, such as the factors that control the size of these compartments.

controlling such co-assembly processes have yet to be established, and it is not clear how to design systems to maximize encapsulation.

In this article we take a step toward this goal, by developing theoretical and computational models that describe the dynamical encapsulation of hundreds of cargo molecules by self-assembling icosahedral shells. Although our models are general, we are motivated by recent experiments on a type of BMC known as the carboxysome (*Kerfeld et al., 2010*; *Schmid et al., 2006*; *Iancu et al., 2007*; *Tanaka et al., 2008*). Carboxysomes are large (40–400 nm), roughly icosahedral shells that encapsulate a dense complex of the enzyme ribulose-1,5-bisphosphate carboxylase/oxygenase (RuBisCO) and other proteins to facilitate the Calvin-Bensen-Bassham cycle in autotrophic bacteria (*Price and Badger, 1991*; *Shively et al., 1973*; *Shively et al., 1973*; *Iancu et al., 2007*; *2010*; *Kerfeld et al., 2010*; *Tanaka et al., 2008*). Recently, striking microscopy experiments visualized $\beta-$carboxysome shells assembling on and budding from procarboxysomes (the condensed complex of RuBisCO and other proteins found in the interior of carboxysomes) (*Cameron et al., 2013*; *Chen et al., 2013*). Genomic analysis suggests that many BMCs with diverse functions assemble via similar pathways (*Cameron et al., 2013*; *Kerfeld and Erbilgin, 2015*). However, the mechanisms of budding and pinch-off to close the shell remain incompletely understood because of the small size and transient nature of assembly intermediates. Moreover, experiments suggest that

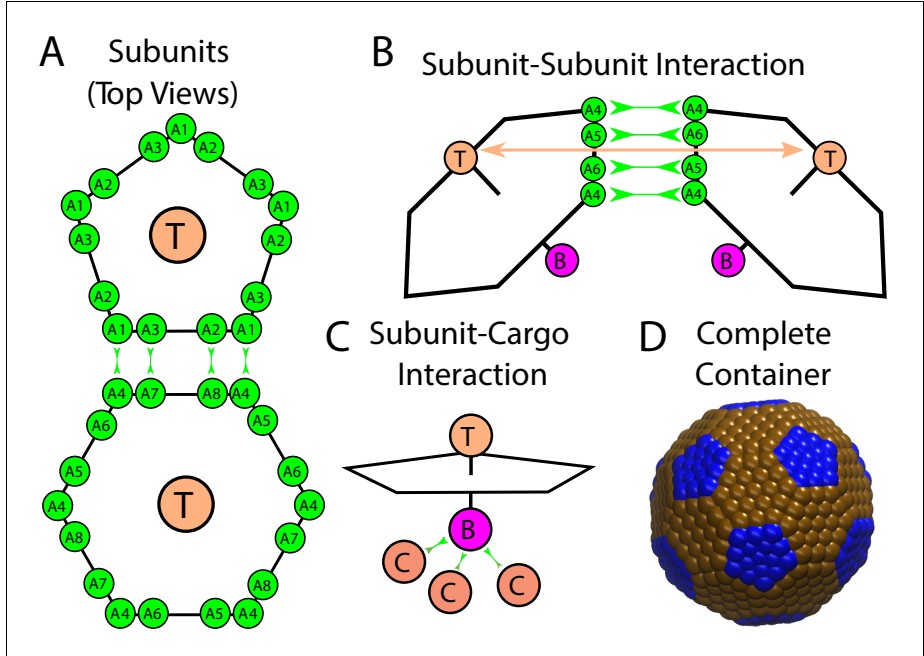

**Figure 1.** Description of the model. (**A**) Each shell subunit contains 'Attractors' (green circles) on the perimeter, a 'Top' (tan circle, 'T') in the center above the plane, and a 'Bottom' (purple circle, 'B' below the plane). (**B**) Interactions between complementary Attractors drive subunit dimerization, with the Top-Top repulsions (tan arrow) tuned to favor the subunit-subunit angle in a complete shell. Complementary pairs of attractors are indicated by green arrows in (**A**) for the pentamer-hexamer interface and in (**B**) for the hexamer-hexamer interface. (**C**) Bottom psuedoatoms bind cargo molecules (terra cotta circles, 'C'), while excluder atoms (blue and brown pseudoatoms in (**D**)) placed in the plane of the pentagon experience excluded volume interactions with the cargo. (**D**) The positions of excluder atoms in the lowest energy shell geometry, a truncated icosahedron with 12 pentamers (blue) and 20 hexamers (brown).

$\alpha-$carboxysomes (another form of carboxysome) assemble by a different mechanism, in which shell assembly encapsulates an initially diffuse pool of RuBisCO (*Iancu et al., 2010*; *Cai et al., 2015*). The factors determining which of these assembly pathways occurs are unknown.

BMC assembly is driven by a complex interplay of interactions among the proteins forming the external shell and the interior cargo. It is difficult, with experiments alone, to parse these interactions for those mechanisms and factors that critically influence assembly pathways, especially due to the lack of an in vitro assembly system. Models which can correlate individual factors to their effect on assembly are therefore an important complement to experiments.

Previous experimental and theoretical studies of encapsulation by icosahedral shells, e.g. the assembly of viral capsids around their nucleic acid genomes (*e.g. Hu and Shklovskii, 2007*; *Kivenson and Hagan, 2010*; *Elrad and Hagan, 2010*; *Perlmutter et al., 2013*; *2014*; *Mahalik and Muthukumar, 2012*; *Zhang et al., 2013*; *Zhang and Linse, 2013*; *Hagan, 2008*; *Devkota et al., 2009*; *Dixit et al., 2006*; *Borodavka et al., 2012*; *Dykeman et al., 2013*; *2014*; *Zlotnick et al., 2013*; *Johnson et al., 2004*; *Patel et al., 2015*; *Cadena-Nava et al., 2012*; *Comas-Garcia et al., 2012*; *2014*; *Garmann et al., 2014a*; *2014b*; *Malyutin and Dragnea, 2013*), have demonstrated that the structure of the cargo can strongly influence assembly pathways and products. However, BMCs assemble around a cargo which is topologically different from a nucleic acid — a fluid complex comprising many, noncovalently linked molecules. We demonstrate here that changing the cargo topology leads to new assembly pathways and different critical control parameters.

We present phase diagrams and analysis of dynamical simulation trajectories showing how the thermodynamics, assembly pathways, and emergent structures depend on the interactions among shell proteins and cargo molecules. Within distinct parameter ranges, we observe two classes of assembly pathways, which resemble those suggested for respectively $\alpha-$ or $\beta-$carboxysomes. We

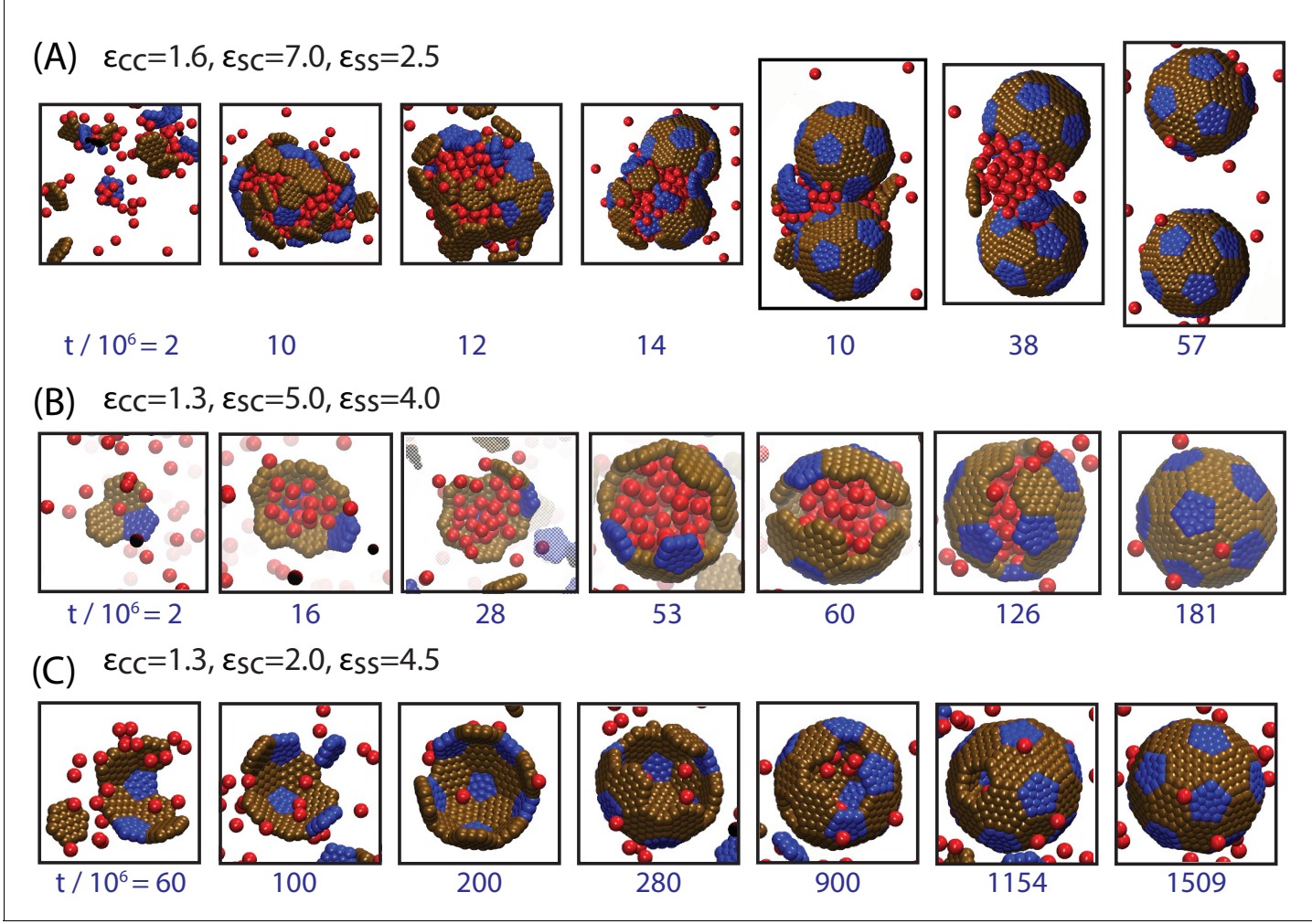

**Figure 2.** Snapshots illustrating typical assembly trajectories. (A) Multi-step assembly involving an amorphous globule of cargo and shell subunits. (B) Single-step assembly, in which shell assembly drives local cargo condensation. and (C) when shell-cargo interactions are too weak to condense the cargo. The values of the cargo-cargo ($\varepsilon_{CC}$), shell subunit-cargo ($\varepsilon_{SC}$), and subunit-subunit ($\varepsilon_{SS}$) interaction strengths are listed above each panel (all energies are in units of the thermal energy $k_BT$), and the time (in units of $10^6$ timesteps) is noted below each image. The color scheme here and throughout the manuscript is: Red=Cargo, Blue=Pentagon Excluder, Brown=Hexagon Excluder. Attractor and Bottom pseudoatoms are omitted to aid visibility. Videos of assembly trajectories are included below.

The following figure supplements are available for figure 2:

**Figure supplement 1.** Snapshots from additional trajectories, including a trajectory with a pre-equilibrated cargo globule.

**Figure supplement 2.** Snapshots from assembly trajectories around anisotropic cargo particles, for (A) strong cargo-cargo interactions leading to two-step, globule-mediated assembly, and (B) weak cargo-cargo interactions leading to simultaneous assembly and cargo condensation.

find that tunability of cargo loading is a key functional difference between the two classes of pathways. Shells assembled around a diffuse cargo can be varied from empty (containing almost no cargo) to completely full, whereas assembly around a condensed, procarboxysome-like complex invariably produces full shells. While we find that the encapsulated cargo becomes ordered due to confinement, complete crystalline order in the globule before encapsulation inhibits budding. We discuss these results in the context of recent observations on carboxysome assembly, and their implications for engineering BMCs, viruses or drug delivery vehicles that assemble around a fluid cargo (*e.g.* Refs. [*Kerfeld and Erbilgin, 2015*; *Parsons et al., 2010*; *Choudhary et al., 2012*; *Lassila et al., 2014*; *Luque et al., 2014*; *Douglas and Young, 1998*; *Rurup et al., 2014*; *Patterson et al., 2014*;

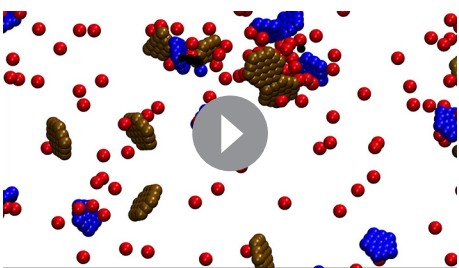

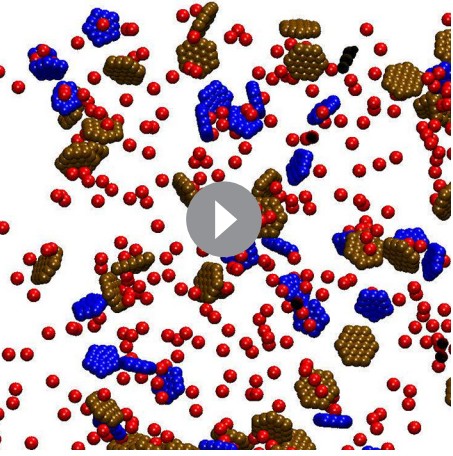

**Video 1.** Animation of a typical simulation showing assembly around a cargo globule. Parameters are $\varepsilon_{CC} = 1.6$, $\varepsilon_{SC} = 7$, and $\varepsilon_{SS} = 2.5$.

*Patterson et al., 2012; Zhu et al., 2014; Rhee et al., 2011; Rurup et al., 2014; Wörsdörfer et al., 2012*]).

**Video 2.** Animation of a typical simulation showing simultaneous assembly and cargo condensation. Parameters are $\varepsilon_{CC} = 1.3$, $\varepsilon_{SC} = 9$, and $\varepsilon_{SS} = 3.5$.

## Results

Our model system is motivated by icosahedral viral capsids and BMCs (*Tanaka et al., 2008; Kerfeld et al., 2010*). Since icosahedral symmetry can accommodate at most 60 identical subunits, formation of large icosahedral structures requires subunits to assemble into different local environments. The subunits can be grouped into pentamers and hexamers, with 12 pentamers at the icosahedron vertices and the remaining subunits in hexamers. Viruses typically assemble from small oligomers of the capsid protein, which we refer to as the basic assembly unit (*Hagan, 2014*). Recent AFM experiments demonstrated that hexamers are the basic assembly unit during the assembly of BMC shell facets (*Sutter et al., 2016*), and the carboxysome major shell proteins crystallize as pentamers and hexamers (*Tanaka et al., 2008*). Motivated by these observations, our model considers two basic assembly units, one a pentamer and the other a hexamer, with interactions designed so that the lowest energy structure corresponds to a truncated icosahedron with 12 pentamers and 20 hexamers (*Figure 1*). While BMCs generally have more hexamers, our model is intended to explore the general principles of assembly around a fluid cargo rather than model a specific system. Further details of the model and a thermodynamic analysis are given in section 3 and the appendices.

To understand how assembly around multiple cargo molecules depends on the relative strengths of interactions between components, we performed dynamical simulations as a function of the parameters controlling shell subunit-subunit ($\varepsilon_{SS}$), shell subunit-cargo ($\varepsilon_{SC}$), and cargo-cargo ($\varepsilon_{CC}$) interaction strengths. All energy values are given in units of the thermal energy, $k_BT$. We focus on parameters for which shell subunit-subunit interactions are too weak to drive assembly in the absence of cargo ($\varepsilon_{SS} \leq 4.5$). Except where mentioned otherwise, the cargo diameter is set equal to the circumradius of a shell subunit.

For the simulated density of cargo particles, the phase behavior (in the absence of shells) corresponds to a vapor at $\varepsilon_{CC} = 1.3$, liquid-vapor phase coexistence for $\varepsilon_{CC} \in [1.6, 2.0]$ (the phase coexistence boundary is slightly below $\varepsilon_{CC} = 1.6$), and a solid phase at $\varepsilon_{CC} = 3.0$. We find that tuning $\varepsilon_{CC}$ through phase coexistence dramatically alters the typical assembly process. Strong cargo interactions ($\varepsilon_{CC} \geq 1.6$) drive formation of a globule followed by assembly and budding of a shell, such as observed for $\beta-$carboxysomes (*Figure 2A*, Simulation *Video 1*), while under weak interactions ($\varepsilon_{CC}<1.6$) shell assembly usually proceeds in concert with cargo encapsulation (*Figure 2B*, Simulation *Video 2*), as suggested for assembly of $\alpha-$carboxysomes. We now elaborate on these classes of assembly pathways, and how the resulting assembly products depend on parameter values.

### Assembly and budding from a cargo globule

We begin by discussing assembly behavior when the cargo-cargo interactions are strong enough to drive equilibrium phase coexistence ($\varepsilon_{CC} \geq 1.6$). Near the phase boundary ($\varepsilon_{CC} = 1.6$) a system of pure cargo particles is metastable on the timescales we simulate. However, for $\varepsilon_{SC}>4$, adding shell

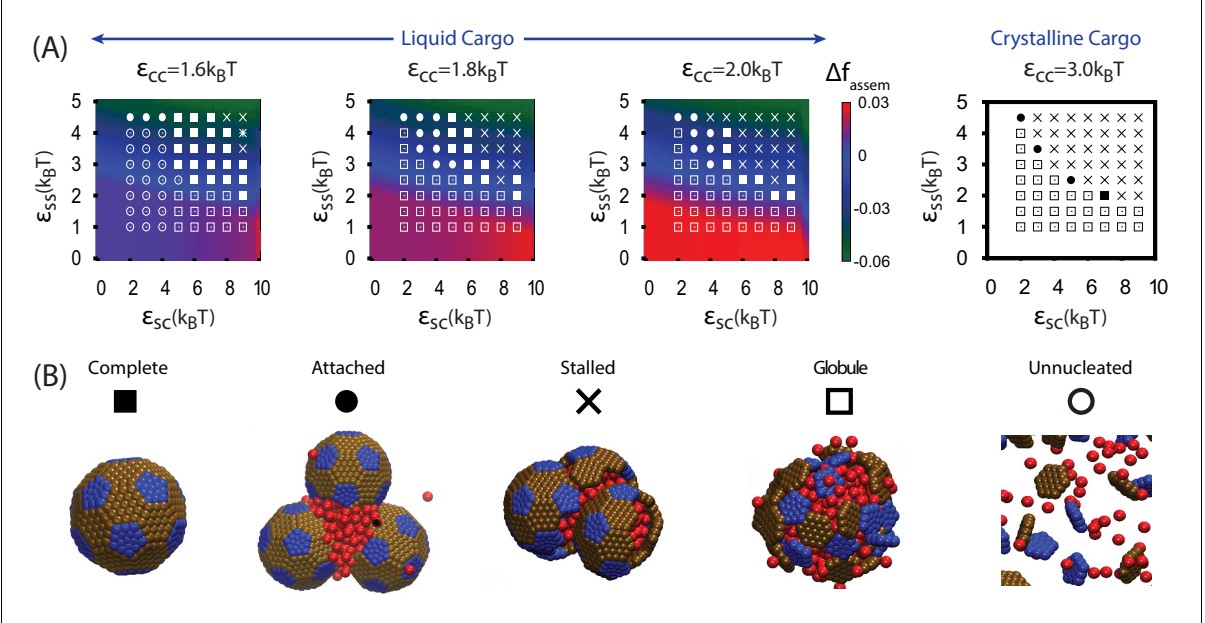

**Figure 3.** Results of assembly around a cargo globule. (A) The most frequently observed assembly outcome is overlaid on a color map of the theoretical free energy density difference $\Delta f_{\mathrm{assem}}$ (*Equation (3)*) between assembled shells and the unassembled globule. Results are plotted against the shell-cargo adsorption strength $\varepsilon_{\mathrm{SC}}$ and the shell-shell interaction strength $\varepsilon_{\mathrm{SS}}$ for indicated values of the cargo-cargo interaction strength $\varepsilon_{\mathrm{CC}}$. (B) Representative snapshots of the predominant assembly outcomes shown in (A).

The following source data and figure supplements are available for figure 3:

**Source data 1.** List of all simulation outcomes for *Figures 3A,5A*.

**Source data 2.** Criteria used to categorize assembly outcomes.

**Figure supplement 1.** The distribution of assembly outcomes in *Figure 3A* is shown as a function of $\varepsilon_{\mathrm{SC}}$ for indicated values of $\varepsilon_{\mathrm{CC}}$ and $\varepsilon_{\mathrm{SS}}$.

**Figure supplement 2.** Results of assembly around a pre-equilibrated cargo globule.

**Figure supplement 2—source data 1.** List of all simulation outcomes for *Figure 3—figure supplement 1—2*.

**Figure supplement 3.** The number of cargo particles packaged as a function of parameters.

subunits drives nucleation of a cargo globule with shell subunits adsorbed on the surface. The subsequent fate of the globule depends on parameter values; typical simulation end-states are shown as a function of parameter values in *Figure 3*. For moderate interaction strengths ($2.5 \leq \varepsilon_{\mathrm{SS}} \leq 3.5$) the globule grows to a large size, typically containing at least twice the cargo molecules that can be packaged within a complete shell. Adsorbed shell subunits then reversibly associate to form ordered clusters. Once a cluster acquires enough inter-subunit interactions to be a stable nucleus, it grows by coagulation of additional subunits or other adsorbed clusters. For the parameter set corresponding to *Figure 2A*, nucleation is fast in comparison to cluster growth, and thus two nuclei grow simultaneously. The last three images show the system immediately preceding and following detachment of the lower shell. Missing only one of its 32 subunits, the shell is connected to the remainder of the droplet only by a narrow neck of cargo. Insertion of the final subunit breaks the neck and completes shell detachment. The complete shell contains 120–130 cargo particles, which is slighty above random close packing ($\approx 120$ particles) but below fcc density ($\approx 150$ particles, see appendix 1.2).

Increasing the shell-shell interaction strength drives faster shell assembly and closure, thus limiting the size of the globule before budding. For the largest interaction strength we simulated ($\varepsilon_{\mathrm{SS}} = 4.5$) the globule typically does not exceed the size of a single shell, and multiple globules nucleate within the simulation box (*Figure 2—figure supplement 1*). This observation could place an upper bound

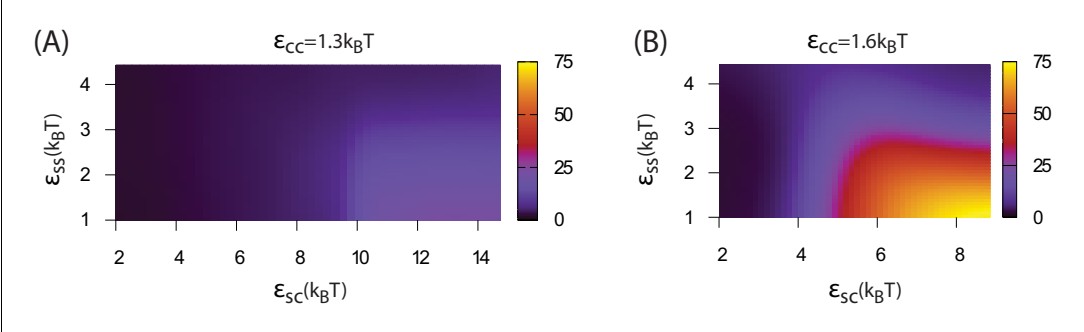

**Figure 4.** Dependence of assembly pathway on shell-cargo and shell-shell interaction strength. The assembly order parameter, defined as the maximum number of unassembled shell subunits adsorbed on a globule at any point during a trajectory, is shown as a function of $\varepsilon_{SC}$ and $\varepsilon_{SS}$ for indicated values of the cargo-cargo interaction $\varepsilon_{CC}$. Large numbers of adsorbed unassembled subunits (>32) indicate the two step assembly mechanism (*Figure 2A*), whereas smaller values correspond to simultaneous assembly and cargo condensation (*Figure 2B*).
The following figure supplement is available for figure 4:

**Figure supplement 1.** Assembly order parameter values for $\varepsilon_{CC} = 1.8$ and $\varepsilon_{CC} = 2.0$.

on shell-shell interaction strengths, since multiple nucleation events were rare in the carboxysome assembly experiments (*Cameron et al., 2013*) (however, we discuss potential complicating factors within the cellular environment below). To quantify the relationship between assembly mechanism and parameter values, we calculate an assembly order parameter, defined as the maximum number of unassembled subunits adsorbed onto a globule during an assembly trajectory. The order parameter is shown as a function of the interaction strengths in *Figure 4*. For $\varepsilon_{CC} \geq 1.6$ and $\varepsilon_{SS} \leq 3$ we observe large values of the order parameter (*e.g.* >32, the red and yellow regions in *Figure 4*), which indicate formation of a large amorphous globule consient with the procarboxysome precursor to carboxysome shell assembly (*Cameron et al., 2013*).

## Other assembly products

Outside of the optimal parameter ranges, we observe several classes of alternative outcomes. Overly weak shell-shell interactions fail to drive assembly. For $\varepsilon_{CC} = 1.6$ and $\varepsilon_{SC} \leq 4$ the cargo vapor phase is metastable, and the system remains 'Unnucleated' (with no cargo globule) on simulated timescales (we discuss alternative initial conditions below). Stronger cargo-cargo or shell-cargo interactions result in unassembled 'Globules', where a cargo globule forms but the shell subunits on its surface fail to nucleate. As $\varepsilon_{SS}$ increases, we observe assembly on the globule, leading either to complete shells or two classes of incomplete assembly. In the first incomplete case, 'Attached', one or more shells almost reaches completion, but fails to detach from the droplet within simulated timescales. 'Attached' configurations occur for low $\varepsilon_{SC}$, when the subunit-cargo interaction does not provide a strong enough driving force for the last subunit(s) to penetrate the cargo and close the shell. Overly strong interactions drive the other class of incomplete assembly: 'Over-nucleated/Malformed', in which an excess of partially assembled shells deplete the system of free subunits before any shells are completed. In this regime it is also common to observe malformed structures, in which defects become trapped within growing shells.

As the cargo-cargo interaction increases ($\varepsilon_{CC} \geq 1.8$), multiple effects narrow the parameter range that leads to complete assembly and detachment. Firstly, cargo globules nucleate rapidly at multiple locations within the simulation box, increasing the likelihood of the 'Over-nucleated' outcome. Secondly, the threshold value of $\varepsilon_{SC}$ required for cargo penetration increases, resulting in 'Attached' shells over a wider parameter range. We also observe a configuration we refer to as 'Stalled', in which shell assembly fails to penetrate the globule surface (and thus does not even proceed to the attached stage). The latter is especially prevalent for $\varepsilon_{CC} = 3.0$, when the cargo crystallizes even in the absence of shell encapsulation. For both 'Attached' and 'Stalled' configurations, regardless of the initial number of nucleation events, we typically observe coarsening into a large globule.

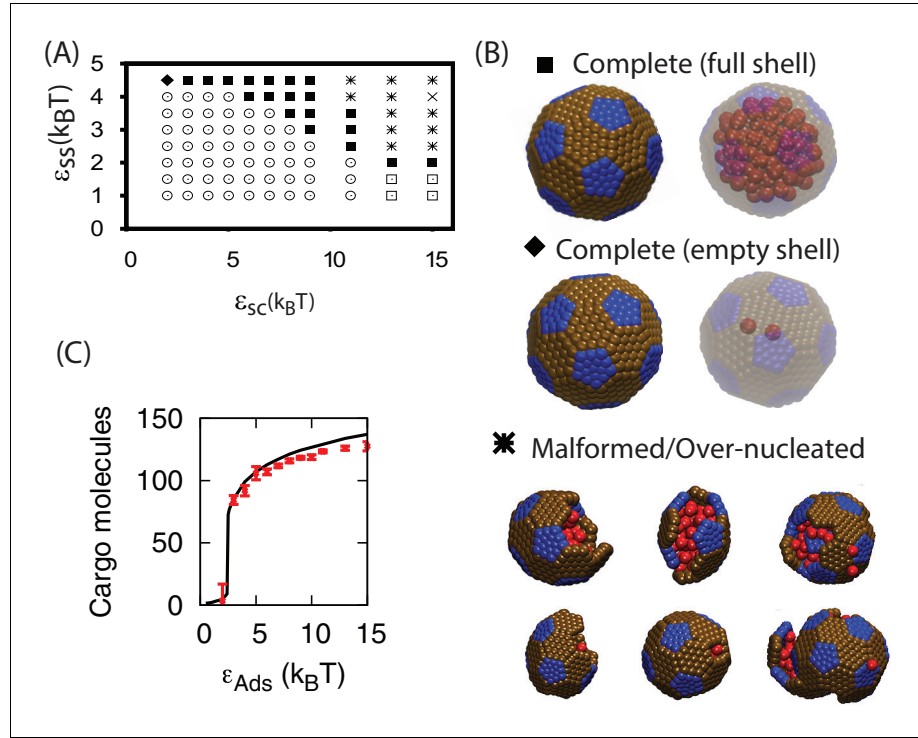

**Figure 5.** Results of assembly around a cargo with weak interactions ($\varepsilon_{CC} = 1.3k_BT$). (**A**) The most frequently observed assembly outcome as a function of $\varepsilon_{SS}$ and $\varepsilon_{SC}$. The distribution of outcomes for $\varepsilon_{SS}$=4 is shown in *Figure 3—figure supplement 2*, and a data file containing the outcome for each trial at each parameter set is included (*Figure 3—source data 1*). (**B**) Representative snapshots for the outcomes shown in (**A**). The complete shell outcomes are shown with the excluders rendered opaque (left) and transparent (right) to enable visualizing the encapsulated cargo. (**C**) The number of cargo molecules encapsulated by shells assembled in dynamics simulations (red symbols) is compared to the results of equilibrium simulations (black line). The dynamics results are averaged over all complete shells (for any $\varepsilon_{SS}$) assembled at each value of $\varepsilon_{SC}$, the error bars indicate 95% confidence intervals. Most simulations were performed for $3 \times 10^8$ timesteps; simulations with $\varepsilon_{SS}$=4.5, $\varepsilon_{SC} \leq 4$, and $\varepsilon_{CC} = 1.3$ exhibited partially assembled shells at $3 \times 10^8$ timesteps, and were continued up to $7.2 \times 10^9$ timesteps.

The following figure supplements are available for figure 5:

**Figure supplement 1.** Assembly yields calculated by simulation and theory.

**Figure supplement 2.** The effect of varying cargo diameter on assembly.

## Simultaneous shell assembly and cargo condensation

For $\varepsilon_{CC} = 1.3$ the cargo forms an equilibrium vapor phase in the absence of shell subunits. However, above threshold values of $\varepsilon_{SS}$ and $\varepsilon_{SC}$, the diffuse cargo molecules drive nucleation of shell assembly. The subsequent assembly pathway depends sensitively on the shell-cargo interaction strength. For low $\varepsilon_{SC}$ (*Figure 2C*), assembly captures only a few cargo molecules, leading to complete, but nearly empty shells. For larger $\varepsilon_{SC}$ (*Figure 2B*, and Simulation *Video 2*), the shell-cargo interactions drive local condensation of cargo molecules. Shell assembly and cargo complexation then proceed in concert, resembling the mechanism proposed for assembly of $\alpha$-carboxysomes (*Iancu et al., 2010*). Thus, tuning the shell-cargo interaction dramatically affects cargo loading, with a sharp transition from empty to filled shells around $\varepsilon_{SC}$=2. This transition closely tracks the equilibrium filling fraction (*Figure 5C*), measured by simulating a complete shell made permeable to cargo molecules. This effect is comparable to the condensation of water vapor below its dew point inside of hydrophilic cavities. In contrast, assembly around a globule only generates full shells.

Assembly of full shells (by either pathway, *Figure 2A* or *Figure 2B*) is typically about two orders of magnitude faster than assembly of empty shells (*Figure 2C*). This disparity demonstrates the key role that the cargo plays in promoting shell association, during all stages of assembly. Cargo molecules initially promote shell nucleation by stabilizing interactions among small, sub-nucleated clusters. Then, the presence of a condensed globule provides a large cross-section for adsorption of additional subunits, significantly enhancing the flux of subunits to the partial capsid, thus increasing its growth rate. The condensed cargo particularly facilitates insertion of the last few subunits, which are significantly hindered by steric interactions, as noted previously for simulations of empty virus capsids (*Nguyen et al., 2007*).

*Figure 5A* shows how the products of assembly around cargo with weak interactions depends on parameters. While moderate parameter values lead to complete assembly, overly weak $\varepsilon_{SC}$ and $\varepsilon_{SS}$ (lower left region of *Figure 5A*) prevent shell nucleation, leading to the 'Unnucleated' outcome. In the limit of large $\varepsilon_{SC}$ but weak $\varepsilon_{SS}$ the shell-cargo interaction stabilizes small disordered globules ($\sim 50$ cargo particles, lower right region of *Figure 5A*), while under strong subunit and weak cargo interactions ($\varepsilon_{SS} = 4.5$, $\varepsilon_{SC} < 5$) shells nucleate but cannot condense the cargo, leading to the complete but slow assembly just discussed. As for assembly around a globule, overly strong interactions lead to overnucleation and malformed shells. However, the predominant mode of malformation is now shell collapse. Because the cargo is below its dew point, the locally condensed globule leads to a negative pressure on the shell subunits, which can flatten the shell and thus prevent closure of a symmetric shell.

## Thermodynamic model

The simple free energy model (*Equations (1–2)*) reproduces the threshold parameter values required for shell assembly with no adjustable parameters (color map in *Figure 3*). Since it is an equilibrium model and only considers the free energy difference between complete and unassembled configurations, it cannot distinguish between parameter values that lead to complete assembly or kinetic traps at the long but finite simulation times. However, the thermodynamic calculation does suggest that the simulations resulting in 'Attached' shells would eventually reach completion on a longer timescale. We do not show $\Delta f_{assem}$ in *Figure 5A* because the globule is always less favorable than assembled shells for $\varepsilon_{CC} = 1.3$, but the yield of well-formed shells in our simulations roughly follows the prediction of the equilibrium theory (*Figure 5—figure supplement 1*).

## Effects of varying other parameters or initial conditions

To investigate whether the results described above depend on assumptions within our model, we performed several sets of additional simulations. Firstly, we performed simulations in which the ratio between cargo diameter in shell subunit size was varied. As shown in *Figure 5—figure supplement 2*, assembly is most robust for our default cargo diameter (for which the model was parameterized), but productive assembly occurs for cargo diameters varied over a factor of four. Secondly, we performed assembly simulations with anisotropic cargo molecules with a shape motivated by the octomer structure of the RuBisCO holoenzyme (*Figure 2—figure supplement 2*).

Thirdly, we performed a set of simulations in which we pre-equilibrated the cargo globule before introducing shell subunits into the system (*Figure 3—figure supplement 2*, Simulation *Video 3*). Investigating this alternative initial condition was motivated by the fact that RuBisCO is present in the cell before induction of the

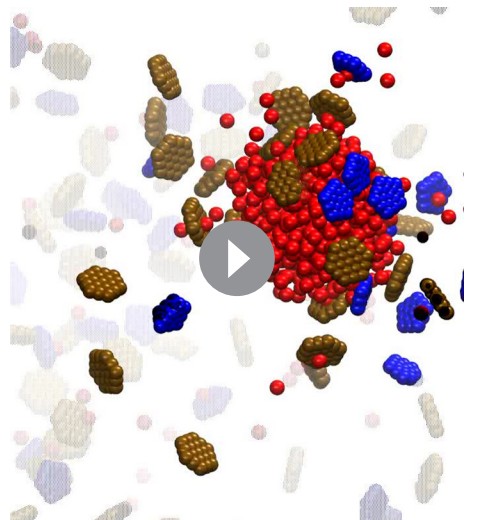

**Video 3.** Animation of a simulation with a pre-equilibrated cargo globule. Parameters are $\varepsilon_{CC} = 1.6$, $\varepsilon_{SC} = 6$, and $\varepsilon_{SS} = 3.5$.

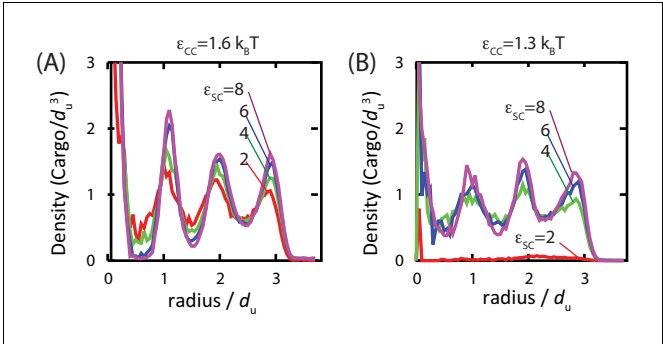

**Figure 6.** Order of the encapsulated cargo. The spherically averaged density of cargo molecules inside a shell is shown as a function of radius for (A) $\varepsilon_{CC}=1.6$ and (B) $\varepsilon_{CC}=1.3$ for indicated values of the cargo-shell adhesion strength $\varepsilon_{SC}$, measured in equilibrium simulations. The density of the encapsulated cargo ranges from below random close packing to near hexagonal close packing density as $\varepsilon_{CC}$ and $\varepsilon_{SC}$ are increased (see *Figure 3—figure supplement 3*). A snapshot of cargo inside the shell is shown in *Figure 5—figure supplement 2*. The raw data for this figure is provided in *Figure 6—source data 1*.

The following source data and figure supplements are available for figure 6:

**Source data 1.** Raw data for *Figure 6*.

**Figure supplement 1.** Ordering of the encapsulated cargo is primarily driven by confinement, not adhesion to the inner surface of the shell.

**Figure supplement 1—source data 1.** Raw data for *Figure 6—figure supplement 1*.

carboxysome gene in the experiments of Ref. (*Cameron et al., 2013*), and by the observation that multiple carboxysomes bud sequentially in time from a single procarboxysome. For $\varepsilon_{CC}=1.6$ the results are very similar to those obtained without pre-equilibrating the cargo. However, for $\varepsilon_{CC}>1.6$, successful assembly and detachment is limited to more narrow ranges of shell-shell and shell-cargo interaction strengths than in *Figure 3*, due to an increased prevalence of 'Attached' and 'Stalled' configurations. The latter are particularly common for $\varepsilon_{CC}=3$, when the cargo forms a hexagonally close packed crystal which strongly resists deformation by shell protein assembly.

Taken together, the results from both assembly protocols (*Figure 3* and *Figure 3—figure supplement 2*) suggest that moderate effective cargo-cargo interactions are most consistent with the observations of shell assembly and budding in Refs. (*Cameron et al., 2013*; *Chen et al., 2013*). Such interactions are strong enough to drive cargo globule formation, but malleable enough to allow shell assembly to deform and eventually sever intra-globule interactions.

## Organization of encapsulated cargo

Studies of assembled carboxysomes report varying degrees of order for the encapsulated cargo, ranging from none to paracrystalline order (*Iancu et al., 2007*; *2010*; *Kaneko et al., 2006*; *Schmid et al., 2006*). We therefore studied the relationship between cargo order and interaction parameters using equilibrium simulations (see *Figure 6* and *Figure 6—figure supplement 1*). Below $\varepsilon_{CC}<3k_BT$, we do not observe true fcc order of the encapsulated cargo. However, for all parameters leading to significant filling, even those well below the cargo liquid-vapor transition, the cargo becomes organized in concentric layers (*Figure 6*). We observe similar cargo organizations within shells which have budded from cargo globules in dynamical simulations. These results demonstrate that ordering of the cargo does not require crystallinity of the initial globule. Moreover, the magnitude of ordering increases with cargo loading, but, for fixed loading, is essentially independent of the cargo-shell interaction strength $\varepsilon_{SC}$. We observe ordering within filled shells due to confinement, even if even if $\varepsilon_{SC}$ is set to 0 (*Figure 6—figure supplement 1*), as previously noted by Iancu *et al.* (*Iancu et al., 2007*).

**Table 1.** Description of the assembly outcomes presented in **Figures 3**,**5**.

| Symbol | Name | Description |
|---|---|---|
| ▪ | Complete shell (full) | Complete shell, full of cargo molecules |
| ◆ | Complete shell (empty)r | Complete shell, almost empty of cargo molecules |
| ● | Attached | Nearly complete shells attached to a globule by a neck of cargo |
| ✳ | Over-nucleated/Malformed | Multiple globules, with incomplete or malformed shells on their surfaces |
| × | Stalled | Large globule with multiple incomplete or malformed shells on its surface |
| □ | Globule | Cargo globule with unassembled shell subunits on its surface |
| ⊙ | Unnucleated | Diffuse subunits and cargo molecules |

## Discussion

We have described an equilibrium theory and a dynamical computational model for the assembly of shells around a fluid cargo. Our simulations show that assembly can proceed by two classes of pathways: (i) a multi-step process in which the cargo forms a dense globule, followed by adsorption, assembly, and budding of shell proteins, or (ii) single-step assembly, with simultaneous aggregation of cargo molecules and shell assembly. This result demonstrates that the minimal interactions included in our model are sufficient to drive both classes of assembly pathways, suggesting that they are a generic feature of assembly around a fluid cargo. Moreover, while we cannot rule out the existence of active mechanisms in biological examples such as carboxysomes, our model demonstrates that the same interactions which drive assembly of shells can also drive budding from and closure around an amorphous globule of cargo.

Our results suggest bounds on the relative strengths of interactions that drive BMC assembly in cells. The decisive control parameter determining the assembly pathway is the cohesive energy between cargo molecules, which could arise through direct cargo-cargo interactions or be mediated by auxiliary proteins (*Cameron et al., 2013*). Relatively weak cargo interactions lead to single-step assembly pathways, while stronger interactions favor formation of the cargo-shell globule. However, the strength of cargo-shell and shell-shell interactions also play a role. Strong shell-shell interactions cause assembly to proceed rapidly during globule formation, limiting the size of the globule. Moreover, if a large globule is already present (*e.g.* due to time-dependent protein concentrations within a cell), strong interactions tend to drive malformed assemblies. We find that an important functional difference between the two classes of assembly pathways is control over the amount of packaged cargo. While the multi-step assembly pathways always generate a shell filled with cargo molecules, shells assembling around a diffuse cargo can be tuned from nearly empty to completely full by controlling the strength of cargo-shell interactions.

These results have implications for reengineering BMCs to encapsulate new core enzymes. Recent works demonstrated that protein cargos can be targeted to BMCs via encapsulation peptides that mediate cargo-shell interactions. However, packaged amounts were much lower than for native core enzymes (*Parsons et al., 2010*; *Choudhary et al., 2012*; *Lassila et al., 2014*). Our simulations show that both cargo-shell and cargo-cargo interactions (direct or mediated) must be controlled to assemble full shells.

We also find that a general equilibrium theory describes the ranges of parameter values for which assembly occurs. However, the dynamical simulations demonstrate that, at finite timescales, there is a rich variety of assembly morphologies. Formation of ordered, full shells requires a delicate balance of cargo-cargo, cargo-shell, and shell-shell interactions, all of which must be on the order $5 - 10k_\text{B}T$. This constraint is consistent with previous studies on viruses and other assembly systems, which found that formation of ordered states requires multiple, cooperative weak interactions between subunits (*Hagan, 2014*; *Whitelam and Jack, 2015*). Outside of optimal parameter regimes, the simulations predict alternative outcomes, ranging from no assembly to various alternative trapped intermediates, with the morphology depending on which interaction is strongest. We find that assembly is least robust to parameter variations when the cargo crystallizes before shell assembly. The assembling shell is unable to deform or penetrate the cargo complex, leading to defect-riddled, non-budded complexes. Within the limits of our simplified model, this observation suggests that

procarboxysome complexes are at least partially fluid prior to successful shell assembly. Moreover, we find that observations of ordered cargo within assembled shells may be explained by packing constraints.

An important limitation of the present study is that the model interactions are specific to the shell geometry shown in *Figure 1* (containing 20 hexamers) because alternating edges on hexagonal subunits have attractive interactions only with pentagonal subunits. In reality BMCs contain many more hexamers (formed from multiple protein sequences) and thus must include a greater range of hexamer-hexamer interactions. Extension of the model to allow for this possibility would allow consideration of two important questions: (1) The mechanism controlling insertion of the 12 pentagons required for a closed shell topology. (2) The relationship between assembly pathway and BMC size polydispersity. In particular, experiments suggest that $\beta$-carboxysomes are more polydisperse than $\alpha$-carboxysomes (*Price and Badger, 1991*; *Shively et al., 1973*; *Shively et al., 1973*; *Iancu et al., 2007*; *2010*; *Kerfeld et al., 2010*; *Tanaka et al., 2008*). We speculate that in the case of assembly around vapor-phase cargo, the size of the assembling shell will be primarily dictated by the preferred shell protein curvature and thus relatively uniform. However, during assembly around a condensed globule, the shell protein interactions could be strained to accommodate a globule which is larger or smaller than the preferred curvature, causing the shell size to depend on a complex balance of intermolecular interaction strengths and variables such as the local RuBisCO concentration.

Our model is minimal, intended to elucidate general principles of assembly around a fluid cargo, and thus may apply to diverse systems including prokaryotic microcompartments, viruses, and engineered delivery vehicles. The predicted trends for how assembly mechanisms and morphologies vary with control parameters can be experimentally tested by microscopy experiments. Such testing will be most straightforward in vitro (*e.g. Luque et al., 2014*; *Douglas and Young, 1998*; *Rurup et al., 2014*; *Patterson et al., 2014*; *Patterson et al., 2012*; *Zhu et al., 2014*; *Rhee et al., 2011*; *Rurup et al., 2014*; *Wörsdörfer et al., 2012*), where subunit-subunit interactions can be tuned by varying solution conditions and the stoichiometries of shell and cargo species can be readily varied. While there is currently no BMC assembly system starting from purified components, our findings can be tested in vivo by mutations which alter known protein binding interfaces, or by altering expression levels of RuBisCO or carboxysome proteins.

We anticipate that our model can serve as a qualitative guide for understanding how such multi-component complexes assemble in natural systems, or to reengineer them for new applications. More broadly, our results demonstrate that the properties of encapsulated cargo, such as its topology, geometry and interaction strengths, strongly influence assembly pathways and morphologies.

# Materials and methods

## Computational model

### Shell subunits

We have adapted a model for virus assembly (*Perlmutter et al., 2013*; *2014*; *Perlmutter and Hagan, 2015a*; *Wales, 2005*; *Fejer et al., 2009*; *Johnston et al., 2010*; *Ruiz-Herrero and Hagan, 2015*) to describe assembly of an icosahedral shell around a fluid cargo. Each subunit contains 'Attractors' on its perimeter that mediate subunit-subunit attractions (as in *Ruiz-Herrero and Hagan, 2015*). Attractor interactions are specific – complementary pairs of Attractors (see *Figure 1A,B* and appendix 1) have short-range interactions (modeled by a Morse potential), whereas non-complementary pairs have no interactions. A repulsive interaction between pairs of 'Top' (type 'T') pseudoatoms favors the correct subunit-subunit angle. The 'Bottom' (type 'B') pseudoatoms mediate short-ranged subunit-cargo attractions (*e.g.* due to interactions with shell 'encapsulation peptides' (*Kinney et al., 2012*; *Cameron et al., 2013*; *Fan et al., 2010*)), represented by a Morse potential. We also add a layer of 'Excluders' in the plane of the 'Top' pseudoatoms, which represent subunit-cargo excluded volume interactions. The strengths of subunit-subunit and subunit-cargo attractions are parameterized by potential well depths $\varepsilon_{SS}$ and $\varepsilon_{SC}$ respectively (appendix 1).

### Cargo

As a minimal representation of globular proteins, the cargo is modeled as spherical particles which interact via an attractive Lennard-Jones (LJ) potential, with well-depth $\varepsilon_{CC}$. The attractions implicitly

model hydrophobic and screened electrostatics interactions between cargo molecules, as well as effective cargo-cargo interactions mediated by auxiliary proteins (*e.g.* the carboxysome protein CcmM (*Cameron et al., 2013*)).

## Simulations

We simulated assembly dynamics using the Langevin dynamics algorithm in HOOMD (a software package that uses GPUs to perform highly efficient dynamics simulations [*Anderson et al., 2008*]) and periodic boundary conditions to represent a bulk system. The subunits are modeled as rigid bodies (*Nguyen et al., 2011*). The simulations were performed using a set of fundamental units (URL. http://codeblue.umich.edu/hoomd-blue/doc/page_units.html), with $1d_{\mathrm{u}}$ defined as the circum-radius of the pentagonal subunit (the cargo diameter is also set to $1\ d_{\mathrm{u}}$). Unless specified otherwise, each simulation contained enough subunits to form four complete shells (48 pentamers and 80 hexamers) and 611 cargo particles (a shell typically encapsulates 120–130 cargo particles) in a cubic box with side length $40d_{\mathrm{u}}$. The simulation time step was $0.001$ in dimensionless time units, and dynamics was performed for $3 \times 10^8$ timesteps unless mentioned otherwise.

We performed two sets of simulations, using different initial conditions. In the first, simulations were initialized by introducing cargo particles and shell subunits simultaneously with random positions and orientations (except avoiding high-energy overlaps). The second set of initial conditions was motivated by the possibility that the cargo globule could form before shell subunits reach sufficient concentrations within the cell to undergo assembly. To model this situation, we pre-equilibrated the cargo by performing a long simulation with only cargo particles present. Shell subunits were then introduced with random positions and orientations (excluding high-energy overlaps). For $\varepsilon_{\mathrm{CC}} \geq 1.6$, the assembly simulations thus began with a cargo globule already present. For $\varepsilon_{\mathrm{CC}} < 1.6$ the two protocols are equivalent, since no globule forms during cargo equilibration.

## Sample sizes

To cover the largest range of parameter space possible given the computational expense associated with each simulation, we performed 5 independent simulations at most parameter sets. To assess statistical error and to estimate the distribution of different assembly outcomes, we performed 10 independent trials for one value of $\varepsilon_{\mathrm{SS}}$ at each value of $\varepsilon_{\mathrm{SC}}$ and $\varepsilon_{\mathrm{CC}}$. We also performed additional simulations at parameter sets for which 5 trials did not result in a majority outcome, or when necessary to obtain better statistics on the number of encapsidated cargo particles. Based on these results, performing additional simulations at other parameter values would not qualitatively change our results. (It would increase the statistical accuracy of estimated boundaries between different outcomes; however, these boundaries correspond to crossovers rather than sharp transitions.)

## Thermodynamics of assembly around a fluid cargo

To complement the finite-time simulations, we have developed a general thermodynamic description of assembly around a fluid cargo. We consider shells composed of species $\alpha = 1, 2, \ldots M$, with $n_\alpha^{\mathrm{shell}}$ subunits of species $\alpha$ in a complete shell, which encapsulates $n_0$ cargo molecules (the index 0 refers to cargo molecules henceforth). Assembly occurs from a dilute solution of cargo molecules with density $\rho_0$, shell subunits with density $\rho_\alpha$ for each species, and the density of assembled, full shells as $\rho_{\mathrm{shell}}$. These are in equilibrium with a globule containing $n_0^{\mathrm{glob}}$ cargo molecules and $n_\alpha^{\mathrm{glob}}$ subunits for each species $\alpha$. We assume that, due to the asymmetric nature of the shell-cargo interaction, the shell subunits reside at the exterior of the globule (as we observe in our simulations). The globule containing unassembled shell subunits thus resembles a spherical microemulsion droplet (*Safran, 1994*). Minimizing the total free energy (see appendix 2) gives:

$$v_0 \rho_{\mathrm{shell}} = \exp\left[ -\left( G_{\mathrm{shell}} - \sum_\alpha n_\alpha^{\mathrm{shell}} \mu_\alpha \right) / k_{\mathrm{B}} T \right] \qquad (1)$$

where $G_{\mathrm{shell}}$ is the interaction free energy of the assembled shell and $\mu_\alpha$ are the chemical potentials of free cargo molecules and shell subunits, given by $\mu_\alpha = k_{\mathrm{B}} T \ln(\rho_\alpha v_0)$, with $v_0$ a standard state volume and the globule composition given by

$$\frac{\partial G_{\text{glob}}(\{n_\alpha^{\text{glob}}\})}{\partial n_\alpha^{\text{glob}}} = \mu_\alpha \quad \text{for } \alpha = 0 \ldots M, \tag{2}$$

with $G_{\text{glob}}(n_{\text{s}}^{\text{glob}}, n_0^{\text{glob}})$ as the globule free energy.

(1) – (2) are the general equilibrium description for a system of assembling shells with a disordered-phase intermediate; application to a specific system requires specifying the forms of $G_{\text{shell}}$ and $G_{\text{glob}}$. In appendix 2 we specify these equations for our computational model, allowing us to compare the equilibrium calculation with simulation results, using no free parameters.

To compare the relative stabilities of the globule and assembled shells, we also calculate the free energy difference

$$\Delta f_{\text{assem}} = f_{\text{tot}}(\{n_\alpha^{\text{glob}} = 0\}) - f_{\text{tot}}(\rho_{\text{shell}} = 0), \tag{3}$$

where the first term on the right-hand side is the minimized free energy for a system containing shells and free subunits but no globule, while the second term corresponds to the minimized free energy for a system containing subunits and the globule, but no assembly.

## Acknowledgements

We are grateful to Maxim Prigozhin for illuminating discussions and for introducing us to the carboxysome assembly problem, and to Fei Cai, Cheryl Kerfeld and Charles Knobler for comments on the manuscript. This work was supported by Award Number R01GM108021 from the National Institute Of General Medical Sciences and the Brandeis Center for Bioinspired Soft Materials, an NSF MRSEC, DMR-1420382. Computational resources were provided by NSF XSEDE computing resources (Maverick and Keeneland) and the Brandeis HPCC which is partially supported by DMR-1420382. MFH performed part of this work while at the Aspen Center for Physics, which is supported by NSF grant PHY-1066293.

## Additional information

### Funding

| Funder | Grant reference number | Author |
|---|---|---|
| National Institute of General Medical Sciences | R01GM108021 | Jason D Perlmutter<br>Farzaneh Mohajerani<br>Michael F Hagan |
| National Science Foundation | DMR-1420382 | Michael F Hagan |

The funders had no role in study design, data collection and interpretation, or the decision to submit the work for publication.

### Author contributions

JDP, MFH, Conception and design, Acquisition of data, Analysis and interpretation of data, Drafting or revising the article; FM, Acquisition of data, Analysis and interpretation of data, Drafting or revising the article

### Author ORCIDs

Michael F Hagan, http://orcid.org/0000-0002-9211-2434

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

## Appendix 1: Model Details

### 1.1 Interaction potentials

Our subunit model is based on a model for viral capsid assembly, developed by Wales (**Wales, 2005**) and Johnston et al. (**Johnston et al., 2010**), which we have adapted to describe interactions with cargo molecules.

Each subunit contains 'Attractors' on its perimeter that mediate subunit-subunit attraction (as in [**Ruiz-Herrero and Hagan, 2015**]). Attractor interactions are specific – complementary pairs of Attractors have short-range interactions (modeled by a Morse potential), whereas non-complementary pairs have no interactions. For simplicity, complementarity is defined based only on the low-energy structure (**Figure 1D**); i.e., there is no attraction between pairs of pentagons. Complementary pairs of attractors are: for the hexagon-hexagon interaction, A4-A4, A5-A6, and for the hexagon-pentagon interaction A1-A4, A2-A8, A3-A7. The strength of attractive interactions is parameterized by the well-depth $\varepsilon_{\mathrm{SS}}$. Because vertex attractors (A1, A4) have multiple partners in an assembled structure, whereas edge attractors have only one, the well-depth for A1-A4 and A4-A4 interactions is set to $\varepsilon_{\mathrm{SS}}/2$, while all other attractor interactions use $\varepsilon_{\mathrm{SS}}$. The 'Top' height, or distance out of the attractor plane, sets the Top-Top distance between interacting subunits, which determines the preferred subunit-subunit angle. We use a height of $h = 1/2r_{\mathrm{b}}$, with $r_{\mathrm{b}} = 1$ the distance between a vertex attractor and the center of the pentagon. The 'Bottom' (type 'B') pseudoatoms mediate subunit-cargo attractions, represented by a Morse potential with well-depth $\varepsilon_{\mathrm{SC}}$. We also add a layer of 'Excluders' in the plane of the 'Top' pseudoatoms (positioned as in **Figure 1**), which represent subunit-cargo excluded volume interactions.

In our model, all potentials can be decomposed into pairwise interactions. Potentials involving container subunits further decompose into pairwise interactions between their constituent building blocks – the excluders, attractors, 'Top', and 'Bottom' pseudoatoms. It is convenient to state the total energy of the system as the sum of three terms, involving subunit-subunit ($U_{\mathrm{SS}}$), cargo-cargo ($U_{\mathrm{LJ}}$), and subunit-cargo ($U_{\mathrm{Ads}}$) interactions, each summed over all pairs of the appropriate type:

$$U = \sum_{\mathrm{sub}\,i}\sum_{\mathrm{sub}\,j<i} U_{\mathrm{SS}} + \sum_{\mathrm{cargo}\,i}\sum_{\mathrm{cargo}\,j<i} U_{\mathrm{LJ}} + \sum_{\mathrm{sub}\,i}\sum_{\mathrm{cargo}\,j} U_{\mathrm{Ads}} \qquad (A1)$$

where $\sum_{\mathrm{sub}\,i}\sum_{\mathrm{sub}\,j<i}$ is the sum over all distinct pairs of subunits in the system, $\sum_{\mathrm{sub}\,i}\sum_{\mathrm{cargo}\,j}$ is the sum over all subunit-cargo particle pairs, etc.

### Subunit-subunit interactions

The subunit-subunit potential $U_{\mathrm{SS}}$ is the sum of the attractive interactions between complementary attractors, and geometry guiding repulsive interactions between 'Top' - 'Top', 'Bottom' - 'Bottom', and 'Top' - 'Bottom' pairs. There are no interactions between members of the same rigid body. Thus, for notational clarity, we index rigid bodies and non-rigid pseudoatoms in Roman, while the pseudoatoms comprising a particular rigid body are indexed in Greek. For subunit $i$ we denote its attractor positions as $\{\mathbf{a}_{i\alpha}\}$ with the set comprising all attractors $\alpha$, its 'Top' position $\{\mathbf{t}_i\}$, and 'Bottom' position $\{\mathbf{b}_i\}$. The subunit-subunit interaction potential between two subunits $i$ and $j$ is then defined as:

$$
\begin{aligned}
U_{\mathrm{SS}}(\{\mathbf{a}_{i\alpha}\}, \mathbf{t}_i, \mathbf{a}_j, \mathbf{t}_j) &= \varepsilon_{\mathrm{SS}} \text{\rlap{Ł}}\left(|\mathbf{t}_i - \mathbf{t}_j|, \sigma_{\mathrm{t},ij}\right) \\
&+ \varepsilon_{\mathrm{SS}} \text{\rlap{Ł}}\left(|\mathbf{b}_i - \mathbf{b}_j|, \sigma_{\mathrm{b}}\right) \\
&+ \varepsilon_{\mathrm{SS}} \text{\rlap{Ł}}\left(|\mathbf{b}_i - \mathbf{t}_j|, \sigma_{\mathrm{tb}}\right) \\
&+ \sum_{\alpha,\beta}^{N_{ai}, N_{aj}} \varepsilon_{\mathrm{SS}} \mathcal{M}\left(|\mathbf{a}_{i\alpha} - \mathbf{a}_{j\beta}|, r_0, \varrho, r_{\mathrm{cut}}^{\mathrm{att}}\right)
\end{aligned}
\tag{A2}
$$

where $\varepsilon_{\mathrm{SS}}$ is an adjustable parameter which both sets the strength of the subunit-subunit attraction at each attractor site and scales the repulsive interactions which enforce the geometry, $N_{ai}$ is the number of attractor pseudoatoms in subunit $i$, $\sigma_{\mathrm{tb}} = 1.8 r_{\mathrm{b}}$ is the diameter of the 'Top' - 'Bottom' interaction (this prevents subunits from binding in inverted configurations (*Johnston et al., 2010*), and $\sigma_{\mathrm{b}} = 1.5 r_{\mathrm{b}}$ is the diameter of the 'Bottom' - 'Bottom' interaction.

In contrast to the latter parameters, $\sigma_{\mathrm{t},ij}$ the effective diameter of the 'Top' - 'Top' interaction, depends on the species of subunits $i$ and $j$; denoting a pentagonal or hexagonal subunit as p or h respectively, $\sigma_{\mathrm{t,pp}} = 2.1 r_{\mathrm{b}}$, $\sigma_{\mathrm{t,hh}} = 2.436 r_{\mathrm{b}}$, and $\sigma_{\mathrm{t,ph}} = 2.269 r_{\mathrm{b}}$. The parameter $r_0$ is the minimum energy attractor distance, set to $0.2 r_{\mathrm{b}}$, $\varrho$ is a parameter determining the width of the attractive interaction, set to $4 r_{\mathrm{b}}$, and $r_{\mathrm{cut}}^{\mathrm{att}}$ is the cutoff distance for the attractor potential set to $2.0 r_{\mathrm{b}}$. Since the interactions just described are sufficient to describe assembly of the shell subunits, we included no excluder-excluder interactions.

The function Ł is defined as the repulsive component of the Lennard-Jones potential shifted to zero at the interaction diameter:

$$
\text{Ł}(x, \sigma) \equiv \theta(\sigma - x) \left[ \left( \frac{\sigma}{x} \right)^{12} - 1 \right]
\tag{A3}
$$

with $\theta(x)$ the Heaviside function. The function $\mathcal{M}$ is a Morse potential:

$$
\begin{aligned}
\mathcal{M}(x, r_0, \varrho, r_{\mathrm{cut}}) &= \theta(r_{\mathrm{cut}} - x) \times \\
&\left[ \left( e^{\varrho\left(1 - \frac{x}{r_0}\right)} - 2 \right) e^{\varrho\left(1 - \frac{x}{r_0}\right)} - V_{\mathrm{shift}}(r_{\mathrm{cut}}) \right]
\end{aligned}
$$

with $V_{\mathrm{shift}}(r_{\mathrm{cut}})$ the value of the potential at $r_{\mathrm{cut}}$.

## Cargo-cargo interactions

The interaction between cargo particles is given by

$$
U_{\mathrm{LJ}}(\{\mathbf{l}_i\}, \{\mathbf{l}_j\}) = \sum_{i<j}^{N_l} \varepsilon_{\mathrm{CC}} \mathcal{L}\left(|\mathbf{l}_i - \mathbf{t}_j|, \sigma_{\mathrm{C}}, r_{\mathrm{cut}}^c\right)
\tag{A5}
$$

with $\mathcal{L}$ the full Lennard-Jones interaction:

$$
\begin{aligned}
\mathcal{L}(x, \sigma, r_{\mathrm{cut}}) &= \theta(x - r_{\mathrm{cut}}) \times \\
&\left\{ 4 \left[ \left( \frac{x}{\sigma} \right)^{12} - \left( \frac{x}{\sigma} \right)^6 \right] - V_{\mathrm{shift}}(r_{\mathrm{cut}}) \right\}
\end{aligned}
\tag{A6}
$$

and $\varepsilon_{\mathrm{CC}}$ is an adjustable parameter which sets the strength of the cargo-cargo interaction, $N_l$ is the number of LJ particles, $\sigma_{\mathrm{C}}$ is the cargo diameter set to $1.0 r_{\mathrm{b}}$ except where mentioned otherwise, and $r_{\mathrm{cut}}^c$ is set to $3\sigma_{\mathrm{C}}$.

## Subunit-cargo interactions

The subunit-cargo interaction is a short-range repulsion between cargo-excluder and cargo-'Top' pairs reresenting the excluded volume plus an attractive interaction between the cargo - 'Bottom' pairs. For subunit $i$ with excluder positions $\{\mathbf{x}_{i\alpha}\}$ and 'Bottom' psuedoatom $\{\mathbf{b}_{i\alpha}\}$ and cargo particle $j$ with position $\mathbf{R}_j$, the potential is:

$$
\begin{aligned}
U_{\mathrm{Ads}}(\{\mathbf{x}_{i\alpha}\}, \mathbf{R}_j) \ &= \sum_{\alpha}^{N_{\mathrm{x}}} \mathcal{L}\big(|\mathbf{x}_{i\alpha} - \mathbf{R}_j|, \sigma_{\mathrm{ex}}\big) \\
&+ \sum_{\alpha}^{N_{\mathrm{t}}} \mathcal{L}\big(|\mathbf{t}_{i\alpha} - \mathbf{R}_j|, \sigma_{\mathrm{t}}\big) \\
&+ \sum_{\alpha}^{N_{\mathrm{b}}} \varepsilon_{\mathrm{SC}} \mathcal{M}\big(|\mathbf{c}_{i\alpha} - \mathbf{R}_j|, \ r_0, \varrho, r_{\mathrm{cut}}\big)
\end{aligned}
$$

where $\varepsilon_{\mathrm{SC}}$ parameterizes the shell-cargo interaction strength, $N_{\mathrm{x}}$, $N_{\mathrm{t}}$, and $N_{\mathrm{b}}$ are the numbers of excluders, 'Top', and 'Bottom' pseudoatoms on a shell subunit, $\sigma_{\mathrm{ex}} = 0.5r_{\mathrm{b}}$ and $\sigma_{\mathrm{t}} = 0.5r_{\mathrm{b}}$ are the effective diameters of the Excluder - cargo and 'Top' - cargo repulsions, $r_0$ is the minimum energy attractor distance, set to $0.5r_{\mathrm{b}}$, $\varrho$ is a parameter determining the width of the attractive interaction, set to $2.5r_{\mathrm{b}}$, and $r_{\mathrm{cut}}$ is the cutoff distance for the attractor potential set to $3.0r_{\mathrm{b}}$.

## Motivation for choice of interaction potentials

The choices we have made for potential functions (Morse or Lennard-Jones) between different classes of pseudoatoms are based on the need for tunability of the interaction length scale and the extent to which guidance on parameterization is available from the existing literature. In particular, the Morse potential enables controlling the interaction length scale independently from the particle excluded volume size, whereas the interaction length scale and excluded volume size are tuned by a single parameter in the Lennard-Jones potential. Our shell-shell interaction potential is based on previous models for viral capsid assembly (*Wales, 2005*; *Johnston et al., 2010*; *Ruiz-Herrero and Hagan, 2015*; *Perlmutter et al., 2013*; *2014*; *Perlmutter and Hagan, 2015b*), and the choice of a Morse potential for attractor-attractor interactions and a Lennard-Jones potential for Top-Top interactions follows these previous works. The attractor interactions are modeled using a Morse potential because the length scale of their interaction strongly affects the subunit orientational specificity. We chose to model the cargo-cargo interaction using a Lennard-Jones potential because the phase behavior for this model has been extensively studied in the literature, thus limiting the need for model parameterization. However, we note that it could be of interest to study how the probability of shell detachment depends on the length scale of the cargo-cargo interaction; we speculate that a longer-range interaction would increase the probability of detachment by making it easier for shell subunits to penetrate into the globule. Finally, the shell-cargo interactions could have used either choice of potential; we elected to use a Morse potential due to its greater flexibility.

## 1.2 Maximum cargo loading

To give context to the densities of packaged cargo particles that we observe in simulations, we estimate the maximum possible cargo loading here. Our assembled shell has the geometry of a truncated icosahedron with an edge length of approximately $1.5d_{\mathrm{u}}$. Accounting for the volume occluded to cargo particles by the shell pseudoatoms, the interior volume is $V_{\mathrm{in}} \approx 109d_{\mathrm{u}}^3$. The maximum number of cargo molecules that can be packaged (assuming hexagonal close packing) is thus $N_{\mathrm{HCP}} \approx 154$. However, this is an overestimate since the shell geometry is not commensurate with perfect hexagonal close packing. We thus estimate $N_{\mathrm{HCP}} = 148$, the

maximum number of packaged cargo particles seen in an equilibrium simulation. The maximum cargo loading for random close packing is then $N_{\mathrm{RCP}} \approx 120$.

## Appendix 2 Thermodynamics of assembly around a fluid cargo

### 2.1 General theory

In this section we present a general thermodynamic description for assembly around a fluid cargo. The theory provides a description of phase behavior in terms of simple physical parameters, and enables evaluating the extent to which our finite-time dynamical simulations have approached equilibrium. We assume that the equilibrium distribution is dominated by three classes of system configurations: free cargo and shell subunits, a disordered globule of cargo molecules with unassembled shell subunits on its surface, and assembled shells filled with cargo molecules. Extension to consider partially assembled intermediates and partially filled shells is straightforward but would complicate the presentation; moreover, at conditions leading to productive assembly, concentrations of partially assembled intermediates are negligible at equilibrium (*Hagan, 2009*; *2014*; *Safran, 1994*; *Gelbart et al., 1994*).

We consider shells composed of species $\alpha = 1, 2, \ldots M$, with $n_\alpha^{\text{shell}}$ subunits of species $\alpha$ in a complete shell, which encapsulates $n_0$ cargo molecules (the index 0 refers to cargo molecules henceforth). Assembly occurs from a dilute solution of cargo molecules with density $\rho_0$, shell subunits with density $\rho_\alpha$ for each species, and the density of assembled, full shells as $\rho_{\text{shell}}$. These are in equilibrium with a globule containing $n_0^{\text{glob}}$ cargo molecules and $n_\alpha^{\text{glob}}$ subunits for each species $\alpha$. The total free energy density is then given by

$$
\begin{aligned}
f_{\text{tot}} \ =& \sum_{\alpha=0}^{M} k_{\text{B}} T \rho_\alpha [\ln(\rho_\alpha v_0) - 1] + k_{\text{B}} T \rho_{\text{shell}} [\ln(\rho_{\text{shell}} v_0) - 1] \\
& + \rho_{\text{shell}} G_{\text{shell}} + V^{-1} G_{\text{glob}} \left( n_0^{\text{glob}}, \{n_\alpha^{\text{glob}}\} \right)
\end{aligned}
\tag{B1}
$$

where the sum runs over free cargo molecules and shell subunits, $V$ is the system volume, $v_0$ is a standard state volume, $G_{\text{shell}}$ is the interaction free energy of the assembled shell, and $G_{\text{glob}}(n_{\text{s}}^{\text{glob}}, n_0^{\text{glob}})$ is the globule free energy. We then minimize $f_{\text{tot}}$ with respect to $N_{\text{shell}} = V \rho_{\text{shell}}$ and $\{n_\alpha^{\text{glob}}\}$, subject to the conservation of mass constraints:

$$
\rho_\alpha^{\text{T}} = \rho_\alpha + n_\alpha^{\text{glob}}/V + \rho_{\text{shell}} n_\alpha^{\text{shell}} \quad \text{for } \alpha = 0 \ldots M
\tag{B2}
$$

where $\rho_\alpha^{\text{T}}$ denotes the total density of species $\alpha$.

This results in *Equations (1–2)* of the main text.

### 2.2 Specification to our computational model

*Equations (1–2)* are the general equilibrium description for a system of assembling shells with a disordered-phase intermediate. To explore how assembly depends on the control parameters ($\varepsilon_{\text{CC}}$, $\varepsilon_{\text{SC}}$, $\varepsilon_{\text{SS}}$, $\rho_{\text{s}}^{\text{T}}$, and $\rho_{\text{s}}^{\text{T}}$) and to compare these equilibrium expressions against our simulation results, we now specify these relations to our computational model.

### 2.2.1 Globule and shell interaction free energies

We model the globule as a liquid droplet of Lennard-Jones (LJ) particles, with shell subunits adsorbed to its exterior surface. For simplicity, we treat shell subunit binding to the globule with the Langmuir adsorption model. To simplify the notation, we suppress dependencies on

control parameters in the free energy expressions, but list them beneath. The free energy of the globule is then given by

$$
\begin{aligned}
G_{\mathrm{glob}} \ (n_{\mathrm{p}}^{\mathrm{glob}}, n_{\mathrm{h}}^{\mathrm{glob}}, n_{0}^{\mathrm{glob}}) = \\
\gamma A_{\mathrm{glob}}(n_{0}^{\mathrm{glob}}) + \mu_{\mathrm{liq}} n_{0}^{\mathrm{glob}} \\
+ g_{\mathrm{Ads}}(n_{\mathrm{p}}^{\mathrm{glob}} + n_{\mathrm{h}}^{\mathrm{glob}}) \\
+ G_{\mathrm{mix}}\left(n_{\mathrm{p}}^{\mathrm{glob}}, n_{\mathrm{h}}^{\mathrm{glob}}, n_{\mathrm{max}}(A_{\mathrm{glob}}(n_{0}^{\mathrm{glob}}))\right),
\end{aligned} \tag{B3}
$$

where $\gamma(\varepsilon_{\mathrm{CC}})$ and $\mu_{\mathrm{liq}}(\varepsilon_{\mathrm{CC}})$ are the bulk surface tension and chemical potential of a LJ liquid, $g_{\mathrm{Ads}}(\varepsilon_{\mathrm{SC}})$ is the shell subunit absorption free energy, $n_{\mathrm{p}}^{\mathrm{glob}}$ and $n_{\mathrm{h}}^{\mathrm{glob}}$ are the numbers of adsorbed pentamers and hexamers respectively, $A_{\mathrm{glob}} = (\sqrt{34}\pi \rho_{\mathrm{liq}}(\varepsilon_{\mathrm{CC}}) n_{0}^{\mathrm{glob}})^{2/3}$ is the area of the globule, and $\rho_{\mathrm{liq}}(\varepsilon_{\mathrm{CC}})$ is the density of the LJ liquid. The final term is the mixing entropy of adsorbed subunits according to Langmuir adsorption, given by

$$
\begin{aligned}
G_{\mathrm{mix}}(n_{\mathrm{p}}^{\mathrm{glob}}, n_{\mathrm{h}}^{\mathrm{glob}}, n_{\mathrm{max}})/k_{\mathrm{B}}T = \\
\ln\left(\dbinom{n_{\mathrm{max}}}{n_{\mathrm{p}}^{\mathrm{glob}}, n_{\mathrm{h}}^{\mathrm{glob}}, n_{\mathrm{max}} - (n_{\mathrm{p}}^{\mathrm{glob}} + n_{\mathrm{h}}^{\mathrm{glob}})}\right),
\end{aligned} \tag{B4}
$$

with $n_{\mathrm{max}}$ as the number of adsorbed subunits at saturation (calculated from simulations, see below).

For the free energy of shell assembly, we consider a shell comprised of $n_{\mathrm{pent}}{=}12$ pentamers and $n_{\mathrm{hex}}$ hexamers, which have $n_{\mathrm{ph}}$ pentamer-hexamer contacts with binding energy $\varepsilon_{\mathrm{ph}}$ and $n_{\mathrm{hh}}$ hexamer-hexamer contacts with energy $\varepsilon_{\mathrm{hh}}$. For our $T{=}3$ model, $n_{\mathrm{hex}}{=}20$, $n_{\mathrm{ph}}{=}60$, and $n_{\mathrm{hh}}{=}30$. The assembly free energy is then given by

$$
\begin{aligned}
G_{\mathrm{shell}} = \ & n_{\mathrm{ph}}\varepsilon_{\mathrm{ph}} + n_{\mathrm{hh}}\varepsilon_{\mathrm{hh}} \\
& - T\left(n_{\mathrm{pent}} s_{\mathrm{pent}} + n_{\mathrm{hex}} s_{\mathrm{hex}} + s_{\mathrm{config}}\right) \\
& + \gamma A_{\mathrm{glob}}(n_{0}^{\mathrm{glob}}) + \mu_{\mathrm{liq}} n_{0}^{\mathrm{glob}} + g_{\mathrm{Ads}}(n_{\mathrm{pent}} + n_{\mathrm{hex}}),
\end{aligned} \tag{B5}
$$

with $s_{\mathrm{pent}}$ and $s_{\mathrm{hex}}$ the translational and rotational entropy penalty associated with binding of pentameric or hexameric subunits and $s_{\mathrm{config}}$ accounting for the configurational entropy associated with subunit and shell symmetries. In our model the pentamers, hexamers, and capsid are 5-fold, 3-fold, and 60-fold symmetric, giving $s_{\mathrm{config}} = k_{\mathrm{B}} \ln(5^{n_{\mathrm{pent}}} 3^{n_{\mathrm{hex}}}/60)$. Other parameters were calculated from simulations, as described next.

## 2.2.2 Determination of parameter values

Since our interactions are constructed from standard potential functions, some of the parameters discussed in the last section are known from the literature, and others can be calculated from simulations. Thus, it is possible to compare our equilibrium theory against simulation results with no fitting parameters. We present the parameter values and how they are obtained in this section.

### Cargo parameters

The parameters characterizing the phase behavior of a Leonard-Jones fluid, $\gamma$, $\mu_{\mathrm{liq}}$, and $\rho_{\mathrm{liq}}$ can be obtained from the literature, but we performed fits specific to the parameter ranges of interest, $1.0 \leq \varepsilon_{\mathrm{CC}}/k_{\mathrm{B}}T \leq 3.0$. The surface tension $\gamma$ was estimated using the approach of Mecke et al. (**Mecke et al., 1997**). We performed separate simulations containing only LJ particles, with numbers of particles and volume for each system set to achieve formation of a

planar liquid vapor interface, and varying values of the LJ interaction strength $\varepsilon_{CC}$. We then calculated $\gamma$ from the virial expression. For our LJ potential, truncated at $r_{cut} = 3\sigma$, we obtain (using the functional form of Ref. (**Mecke et al., 1997**)

$$\gamma(\varepsilon_{CC}) = 2.936 \left( 1 - \frac{\varepsilon_{CC}^{-1}}{1.3} \right)^{1.688}.$$ (B6)

From the same simulations, we calculated the dependence of the bulk liquid density on $\varepsilon_{CC}$ as

$$\rho_{liq}(\varepsilon_{CC}) = -1.439 + 2.165\varepsilon_{CC}^{0.115}.$$ (B7)

Although there are a number of empirical forms for the LJ equation of state available in the literature, they vary widely in complexity, number of fit parameters, and presumably accuracy over the parameter range we are interested in. We therefore estimated the liquid chemical potential $\mu_{liq}$ from the vapor-phase densities $\rho_{vap}$ in LJ liquid-vapor coexistence simulations according to

$$\mu_{liq} = k_B T \ln(\rho_{vap}\sigma^3) - A\gamma/N_{liq},$$ (B8)

where $A$ is the interfacial area, $N_{liq}$ is the total number of particles in the liquid phase as a function of $\varepsilon_{CC}$, and $\gamma$ is given by **Equation B6**. The results are fit well by the linear function

$$\mu_{liq}(\varepsilon_{CC}) = 3.13 k_B T - 5.6\varepsilon_{CC}.$$ (B9)

## Shell subunit-subunit interactions

We estimated the subunit-subunit binding free energy values as functions of the well-depth parameter $\varepsilon_{SS}$ by measuring the dimerization equilibrium constant in simulations of subunits only capable of forming dimers (**Figure 1C**). For both pentamer-hexamer and hexamer-hexamer dimers, we obtain binding free energies which are linear functions of the well-depth $\varepsilon_{SS}$. We interpret the y-intercept as the binding entropy, giving:

$$
\begin{aligned}
g_{ph} &= \varepsilon_{ph}\varepsilon_{SS} - Ts_{pent} \\
\varepsilon_{ph} &= -2.95; \quad s_{pent} = -17.2k_B \\
g_{hh} &= \varepsilon_{hh}\varepsilon_{SS} - Ts_{hex} \\
\varepsilon_{hh} &= -3.15; \quad s_{hex} = -17.7k_B
\end{aligned}
$$

where the standard state volume is $d_u^3$.

In **Equation (B5)** we then make the assumption that, because the interactions are orientationally specific, a subunit incurs its entire binding entropy penalty upon dimerization — because a bound subunit is already aligned to form additional interactions, these interactions do not lead to further entropy penalties. In reality, this is an under-prediction since some additional entropy losses occur on making additional bonds (**Hagan and Chandler, 2006**; **Hagan et al., 2011**), but these are not sufficiently large to qualitatively affect our results.

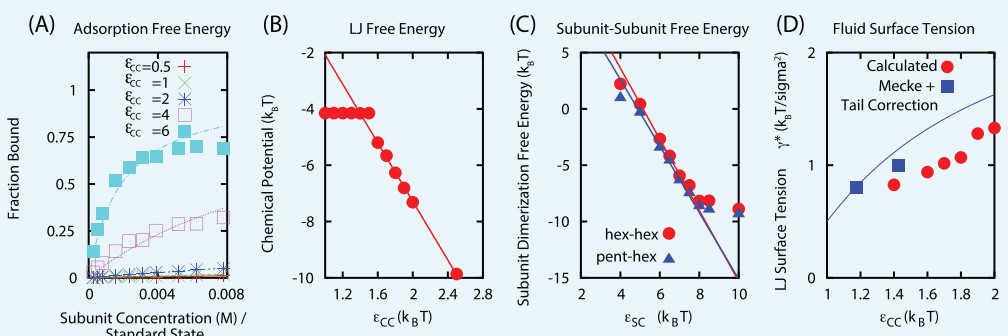

**Appendix 2—figure 1.** (**A**) Langmuir isotherms to estimate $g_{\mathrm{Ads}}(\varepsilon_{\mathrm{SC}})$. (**B**) Estimate of the chemical potential for an equilibrated LJ system (before correcting for the finite size of liquid droplet). (**C**) Fit of the subunit dimerization free energies $g_{\mathrm{hh}}(\varepsilon_{\mathrm{SS}})$ and $g_{\mathrm{ph}}(\varepsilon_{\mathrm{SS}})$ as a function of the well depth parameter $\varepsilon_{\mathrm{SS}}$. (**D**) Fit of LJ droplet surface tension, including the tail correction. DOI: 10.7554/eLife.14078.027

## Shell subunit adsorption onto globule

We estimated the shell subunit adsorption free energy by performing simulations of subunits which cannot assemble ($\varepsilon_{\mathrm{SS}}=0$) in the presence of a cargo globule. We then measured the globule size and number of adsorbed subunits as functions of $\varepsilon_{\mathrm{SC}}$. We found the results could be fit using the Langmuir adsorption model, with the adsorption free energy of a single subunit $g_{\mathrm{Ads}}$ as a fit parameter for each value of $\varepsilon_{\mathrm{SC}}$. We assumed that the maximum number of adsorbed subunits (the number of lattice sites in the Langmuir model) does not directly depend on $\varepsilon_{\mathrm{SC}}$, and hence fit this parameter globally, obtaining $n_{\mathrm{max}}=80$ for a globule with $n_0^{\mathrm{glob}}=300$ cargo molecules. In our calculations we assume that $n_{\mathrm{max}}$ is proportional to the globule surface area, consistent with observations from simulations. Our fit resulted in a linear relationship between the adsorption energy and free energy over the range of interest:

$$g_{\mathrm{Ads}} = 0.093k_{\mathrm{B}}T - 1.17\varepsilon_{\mathrm{SC}}. \tag{B10}$$

