## [Decision Letter]

Thank you for submitting your work entitled "Many-molecule encapsulation by an icosahedral shell" for consideration by *eLife*. Your article has been reviewed by three peer reviewers, and the evaluation has been overseen by a Reviewing Editor and Naama Barkai as the Senior Editor.

The reviewers have discussed the reviews with one another and the Reviewing Editor has drafted this decision to help you prepare a revised submission.

The following individuals involved in review of your submission have agreed to reveal their identity: Charles Knobler and Avinoam Ben-Shau (peer reviewers).

The manuscript describes novel simulations of encapsulation of hundreds of cargo molecules in icosahedral shells. This is relevant for the formation of the carboxysome microcompartment in bacteria. The work is very impressive from a theoretical standpoint and the tendency is to accept the paper for publication in *eLife*. However, following the main comment of Reviewer 1, re-enforced by Reviewer 3, there was debate about the suitability of the work for the biological audience of *eLife*. The overall feeling is that it is, but we suggest that the relevance to biology will be further emphasized. In producing the final draft of the manuscript the authors should also address the following issues:

1) Resolve the confusion about the assembly outcome (see comment of Reviewer 1).

2) Refer to Reviewer 1's point regarding the monodispersity differences between α- and β-carboxysomes.

3) Examine the effect of changing the diameter of the cargo on the results (Reviewers 1 and 2).

4) Explain why the cargo is allowed to equilibrate prior the addition of the shell, in contrast with the biological scenario (Reviewer 1).

5) Consider changing the names of the energy terms (Reviewer 1).

6) Explain why a Morse potential is used for the shell-shell and shell-cargo interactions while a Lennard-Jones potential is used for cargo-cargo interactions (Reviewer 1).

7) Consider adding the effect of non-isotropic interactions among the cargo molecules (Reviewer 1).

8) Cite the Sutter et al. article.

9) Address the point about making the model geometry more permissive (Reviewer 1).

10) Address the issue about Figure 4 (Reviewer 1).

11) Consider adding the analogy to the wetting behavior of binary, immiscible, liquid droplets but not at the expense of making the paper less clear to biologists (Reviewer 3).

Reviewer #1:

Bacterial microcompartments (BMCs) are capsid-like structures that function as organelles in bacteria. They are composed of both shell proteins and specific cargo proteins that self-assemble into the ~100 nm structures found in bacterial cells. Recent experimental evidence suggests that different BMCs may assemble through different mechanisms. Here, Perlmutter provide a compelling theoretical evidence of such mechanisms. The authors demonstrate that this minimal set of interactions is sufficient to drive efficient encapsulation, and they explore the phase diagrams with respect to these parameters. Two distinct paths to well-formed full shells are identified which are consistent with current hypotheses for α- and β-carboxysome assembly, respectively. The balance of energetics they identify could provide a useful conceptual framework for future work on in vitro reconstitution of existing BMCs and the engineering of synthetic ones. The manuscript is well written, and the work appears to be competently executed and analyzed.

My main concern is about the suitability of this paper for *eLife* and its readership. With its emphasis on statistical mechanical methods and interpretations it seems like this would be a better fit in something like the Biophysical Journal or J. Phys. Chem. B. Moreover, the work is likely to benefit from the exposure in one of these specific journals, as opposed to the more biological readership of *eLife*.

I have another general concern about the categorization of assembly outcomes, which I found confusing. For instance in Figure 3 'unnucleated' is shown as a mass of cargo and shell pieces whereas in Figure 5 'unnucleated' is a collection of separated pieces. It is also not clear to me what exactly 'vapor assembly' means because the section "Vapor Assembly" describes the pathway through which α-carboxysomes are thought to assemble. Are those complete shells filled through this way counted as 'complete' or 'vapor assembly' and what determines the cutoff, the amount of cargo? There also seems to be inconsistent use of symbols where X indicates 'Stalled' in Figure 3 but 'Malformed' in Figure 5. The Results needs a clear and organized description of each of the states as well as consistent usage of images and symbols. A table might be a good idea.

Finally, I wonder if the authors have considered the monodispersity differences between α- and β-carboxysomes. If I recall correctly, β-carboxysomes tend to be more heterogeneous between related species and even within the same cell, suggesting some fundamental difference. Does the current model provide any insight into this?

Reviewer #2:

Mike Hagan and his collaborators have carried out what to my mind are the best computer studies of viral assembly by using minimal models that capture the essential physics of the assembly process. Here they carry out a computational; study that addresses a related problem, the assembly of carboxysomes. The model and computational methods they employ are similar to those they have utilized in previous studies but with an additional degree of complexity. In all of their previous work, I believe, they have used models for the protein shell that is appropriate to a T =1 capsid, which can be made up of rigid capsomers of a single type, either triangles or pentamers. Here they employ a model that is appropriate to a T = 3 shell that is made up of rigid pentamers and hexamers. While the interactions between subunits and between subunits and cargo are modeled in a similar fashion in all of these simulations, the introduction of two different subunits considerably complicates the computations. To my untutored understanding of the nuts and bolts of computer modeling, this seems to me to be a tour de force.

The work breaks new ground as well in considering the effects of the phase behavior of the cargo on the assembly path. While the results are very interesting, there are no surprises. The way in which the assembly pathway depends on the strengths of the interactions is what one would have been expected after being informed by the previous computational studies and in vitro assembly studies of viral assembly. They are by no means trivial, however and especially useful because there are no in vitro studies of carboxysome assembly. I think that this is an excellent paper.

Reviewer #3:

In this paper Perlmutter et al. present an interesting computational study of a self-assembly process whereby many small (spherical) cargo particles are encapsulated by icosahedral shells composed of 12 pentagons and 20 hexagons, analogous to the T=3 protein shells of viral capsids. Using Brownian dynamics simulations the authors analyze the assembly ("phase") behavior of a mixture enabling the formation of up to four complete carboxysome-like particles (CLP). The simulations are accompanied by an equilibrium thermodynamic theoretical analysis. Three interaction parameters dictate the assembly behavior of the system ϵ(LJ), ϵ(sub), ϵ(Ads) representing the (magnitudes) of the interaction energies between neighboring cargo units, capsid subunits, and subunit-cargo pairs, respectively. As expected, large ϵ(LJ) leads to solidification of the cargo particles, preventing CLP formation. Similarly large ϵsub (not simulated) would lead to the formation of empty capsids. The interesting regime is when all energies are comparable, not too small and not too large (compared to kT), in which case two scenarios are possible: (i) co-assembly of the cargo and shell and (ii) aggregation of the cargo particles, followed by "carving" their aggregates to form a complete CLPs (provided subunit interactions are strong enough) – as illustrated in the relevant cases in Figure 2,Figure 3, and convincingly summarized in the Discussion.

Reading the Introduction and the Discussion I conclude that the simulations and their analysis are of considerable biological relevance and interest. However, not being a biologist myself, I cannot judge to what extent. On the other hand, as far as the simulations are concerned, my impression is that they are highly non-trivial and of high quality, of the same high standards as the simulations of viral assembly that Hagan et al. have pioneered. The complementing thermodynamic analysis adds a significant theoretical aspect to this work.

Reading the manuscript it also reminded me the (indeed, incomplete) analogy to the wetting behavior of binary, immiscible, liquid droplets. For such droplets of A and B molecules, depending on the interaction strengths ϵ(AA), ϵ(BB), ϵ(AB) complete segregation takes place when (ϵ(AA)+ ϵ(BB))/2>ϵ(AB), and ϵ(AA)> ϵ(AB), ϵ(BB)> ϵ(AB), and envelopment of A by B takes place when (ϵ(AA)+ ϵ(BB))/2>ϵ(AB) > ϵ(BB), etc. The analogy is incomplete because of the special symmetry of the subunit-subunit and the anisotropic subunit-cargo interactions, but some similarity exists and may worth being pointed out. There are few other points which could help the reader (and may already be present in the manuscript but I missed).For instance: we are told that there are 611 cargo particles of a given diameter and we know the inner volume of the capsids. It would be of interest to know what is the maximal number of close-packed spheres can be accommodated within the capsid's volume, and what is the actual density observed. One of the figures shows layering of the cargo particles – at what density relative to the maximal density in the capsid (or in fcc solid) this happens? Other numbers of interest – again, perhaps mentioned – are the critical density and interaction energy of the cargo spheres. Maybe a list of all the "limiting values" of this kind should be mentioned at the beginning of the Results section.

I wonder whether all 78 references are really needed. My guess is that the list can be more economical.

But – to summarize. I find this paper interesting, original and of high scientific quality and recommend its publication in *eLife*.

---

## [Author Response]

*1) Resolve the confusion about the assembly outcome (see comment of Reviewer 1).*

*2) Refer to Reviewer 1's point regarding the monodispersity differences between α- and β-carboxysomes.*

*3) Examine the effect of changing the diameter of the cargo on the results (Reviewers 1 and 2).*

*4) Explain why the cargo is allowed to equilibrate prior the addition of the shell, in contrast with the biological scenario (Reviewer 1).*

*5) Consider changing the names of the energy terms (Reviewer 1).*

*6) Explain why a Morse potential is used for the shell-shell and shell-cargo interactions while a Lennard-Jones potential is used for cargo-cargo interactions (Reviewer 1).*

*7) Consider adding the effect of non-isotropic interactions among the cargo molecules (Reviewer 1).*

*8) Cite the Sutter et al. article.*

*9) Address the point about making the model geometry more permissive (Reviewer 1).*

*10) Address the issue about Figure 4 (Reviewer 1).*

*11) Consider adding the analogy to the wetting behavior of binary, immiscible, liquid droplets but not at the expense of making the paper less clear to biologists (Reviewer 3).*

*12) Address the typos and minor points raised by reviewers 2 and 3.*

We have revised our attached manuscript as suggested by the reviewers. We appreciate both the positive evaluations of our manuscript and the many thoughtful suggestions for its improvement by Prof. Knobler, Prof. Ben-Shaul and the anonymous reviewer. We believe that addressing these comments has significantly improved our presentation. It was clear from the reviewer comments that, given the complex variety of structures that can assemble in the system, we needed to do a better job of clearly presenting our assembly outcomes. We have reformatted our figures to address this. Secondly, there were a number of suggestions for interesting additional simulations (indeed many of these were already on our list of things to do). We have performed these simulations to the extent possible within the two months resubmission time table, leading to several new figures.

*Reviewer #1:*

*Bacterial microcompartments (BMCs) are capsid-like structures that function as organelles in bacteria. They are composed of both shell proteins and specific cargo proteins that self-assemble into the ~100 nm structures found in bacterial cells. Recent experimental evidence suggests that different BMCs may assemble through different mechanisms. Here, Perlmutter provide a compelling theoretical evidence of such mechanisms. The authors demonstrate that this minimal set of interactions is sufficient to drive efficient encapsulation, and they explore the phase diagrams with respect to these parameters. Two distinct paths to well-formed full shells are identified which are consistent with current hypotheses for α- and β-carboxysome assembly, respectively. The balance of energetics they identify could provide a useful conceptual framework for future work on in vitro reconstitution of existing BMCs and the engineering of synthetic ones. The manuscript is well written, and the work appears to be competently executed and analyzed.*

My main concern is about the suitability of this paper for eLife and its readership. With its emphasis on statistical mechanical methods and interpretations it seems like this would be a better fit in something like the Biophysical Journal or J. Phys. Chem. B. Moreover, the work is likely to benefit from the exposure in one of these specific journals, as opposed to the more biological readership of eLife.

While we did initially consider sending our work to a more physical sciences oriented journal, such as JACS or PRL, we felt that we would get more readership from the biology community in *eLife*. It has been our experience that the biology community is interested in and very capable of understanding statistical mechanics arguments when they are properly presented. Moreover, we are particularly interested in having a conversation with the biology community about the questions that our paper addresses, and *eLife* is an excellent platform for this.

We have revised the text in several ways to clarify the biological relevance of our study. In the introduction we now point out that, in a system as complex as BMCs, quantitative models are an essential complement to experiments. This is especially true given the lack of a complete in vitro assembly system. In the results we now explain the relationship between the biological scenario of BMC assembly and the simulation protocols we have used. We have also used an additional protocol (see below) which may connect better to the cellular situation. In the Discussion we point out aspects of our results which set bounds on the relative strengths of interactions that can drive BMC assembly in cells. We also point out limitations of the current model which should be addressed in the future to more completely describe the biological situation, such as the question of size polydispersity.

*I have another general concern about the categorization of assembly outcomes, which I found confusing. For instance in Figure 3 'unnucleated' is shown as a mass of cargo and shell pieces whereas in Figure 5 'unnucleated' is a collection of separated pieces. It is also not clear to me what exactly 'vapor assembly' means because the section "Vapor Assembly" describes the pathway through which α-carboxysomes are thought to assemble. Are those complete shells filled through this way counted as 'complete' or 'vapor assembly' and what determines the cutoff, the amount of cargo? There also seems to be inconsistent use of symbols where X indicates 'Stalled' in Figure 3 but 'Malformed' in Figure 5. The Results needs a clear and organized description of each of the states as well as consistent usage of images and symbols. A table might be a good idea.*

It is apparent from this comment and those of the other reviewers that the rich variety of configurations which assemble in this system was not adequately categorized and illustrated by our set of snapshots. We have now implemented a new, hopefully clearer, set of category names, we use a unique symbol for each different category across all figures, and we have added a table giving a short description for each category. We have also changed the category name that was formerly ‘vapor assembly’ to avoid confusion with the other assembly pathway.

*Finally, I wonder if the authors have considered the monodispersity differences between α- and β-carboxysomes. If I recall correctly, β-carboxysomes tend to be more heterogeneous between related species and even within the same cell, suggesting some fundamental difference. Does the current model provide any insight into this?*

We do think that the different assembly pathways between α- and β- carboxysomes play a role in the size-dispersity of carboxysomes. In the case of assembly around vapor-phase cargo, the size of the assembling carboxysome will be primarily dictated by the preferred carboxysome shell protein curvature. However, during assembly around a pre-formed globule, the shell protein interactions could be strained to accommodate a globule which is larger or smaller than the preferred curvature. The latter affect depends on the protein mechanics and orientational specificity of their interactions. For the present study, in order to focus on the simplest possible case, we designed model interactions which are size-specific and thus suppress polymorphism. Polymorphism is still possible in our model, but likely requires larger strain to the interactions than would be the case for more promiscuous interactions. We are currently investigating the best way to extend the model to allow for greater polymorphism. This requires significant exploration and model space however, and is thus beyond the scope of our current manuscript. We have added a paragraph in the Discussion mentioning the potential link between assembly pathway and size-polydispersity, and the caveat that the specificity of our current model design limits polydispersity.

*Reviewer #3:*

*In this paper Perlmutter et al. present an interesting computational study of a self-assembly process whereby many small (spherical) cargo particles are encapsulated by icosahedral shells composed of 12 pentagons and 20 hexagons, analogous to the T=3 protein shells of viral capsids. Using Brownian dynamics simulations the authors analyze the assembly ("phase") behavior of a mixture enabling the formation of up to four complete carboxysome-like particles (CLP).*

[…]

*Reading the manuscript it also reminded me the (indeed, incomplete) analogy to the wetting behavior of binary, immiscible, liquid droplets. For such droplets of A and B molecules, depending on the interaction strengths ϵ(AA), ϵ(BB), ϵ(AB) complete segregation takes place when (ϵ(AA)+ ϵ(BB))/2>ϵ(AB), and ϵ(AA)> ϵ(AB), ϵ(BB)> ϵ(AB), and envelopment of A by B takes place when (ϵ(AA)+ ϵ(BB))/2>ϵ(AB) > ϵ(BB), etc. The analogy is incomplete because of the special symmetry of the subunit-subunit and the anisotropic subunit-cargo interactions, but some similarity exists and may worth being pointed out.*

The reviewer raises an interesting point here about the role of interaction asymmetry versus like-like/likenonlike interaction strength disparity, which is likely to be of interest to physical scientists familiar with the statistical mechanics of binary mixtures. However, we do fear that this Discussion might be distracting to some biologists who are less familiar with these ideas. As a compromise, we have decided to point out the similarity between a globule with unassembled shell subunits on its surface and surfactants stabilizing a spherical microemulsion droplet, where the asymmetry of the surfactant molecules plays a key role in driving them to the interface. This is in the subheading “Thermodynamics of assembly around a fluid cargo” of the Model Section.

*There are few other points which could help the reader (and may already be present in the manuscript but I missed).For instance: we are told that there are 611 cargo particles of a given diameter and we know the inner volume of the capsids. It would be of interest to know what is the maximal number of close-packed spheres can be accommodated within the capsid's volume, and what is the actual density observed. One of the figures shows layering of the cargo particles – at what density relative to the maximal density in the capsid (or in fcc solid) this happens? Other numbers of interest – again, perhaps mentioned – are the critical density and interaction energy of the cargo spheres. Maybe a list of all the "limiting values" of this kind should be mentioned at the beginning of the Results section.*

We list the location of the binodal as well as the phase behavior corresponding to each of the simulated values of ϵ_CC_ at the beginning of Results. We have now added a section to the appendix in which we estimate the number of packaged cargo particles corresponding to fcc or rcp packing. We have added a plot showing the number of packaged cargo particles as a function of parameter values (Figure 3—figure supplement 3), and Figure 5 also shows the number of packaged cargo particles for vapor phase assembly. As suggested by the reviewer, we now refer to these plots in Figure 4, so that the reader is able to place the observation of layering in the context of packing density. (We observe layering even below rcp density, but, as would be expected, the degree of ordering increases as the packaged cargo approaches fcc density.)

*I wonder whether all 78 references are really needed. My guess is that the list can be more economical.*

There is an extensive background literature both in the field of BMCs and in relating modeling efforts. We feel it is important to point to these works, both for the benefit of the reader who is unfamiliar with BMCs and to acknowledge previous work.